# Loss of the branched-chain amino acid transporter CD98hc alters the development of colonic macrophages in mice

Philipp Wuggenig[1], Berna Kaya[1], Hassan Melhem[1], C. Korcan Ayata[1], Swiss IBD Cohort Investigators*, Petr Hruz[2], A. Emre Sayan [3], Hideki Tsumura[4], Morihiro Ito[5], Julien Roux [1,6] & Jan Hendrik Niess[1,2✉]

Comprehensive development is critical for gut macrophages being essential for the intestinal immune system. However, the underlying mechanisms of macrophage development in the colon remain elusive. To investigate the function of branched-chain amino acids in the development of gut macrophages, an inducible knock-out mouse model for the branched-chain amino acid transporter CD98hc in CX3CR1[+] macrophages was generated. The relatively selective deletion of CD98hc in macrophage populations leads to attenuated severity of chemically-induced colitis that we assessed by clinical, endoscopic, and histological scoring. Single-cell RNA sequencing of colonic lamina propria macrophages revealed that conditional deletion of CD98hc alters the "monocyte waterfall"-development to MHC II[+] macrophages. The change in the macrophage development after deletion of CD98hc is associated with increased apoptotic gene expression. Our results show that CD98hc deletion changes the development of colonic macrophages.

[1] Department of Biomedicine, University of Basel, Basel, Switzerland. [2] University Center for Gastrointestinal and Liver Diseases, St. Clara Hospital and University Hospital, Basel, Switzerland. [3] Cancer Sciences Division, Somers Cancer Research Building, Southampton University, Southampton, UK. [4] Division of Laboratory Animal Resources, Nation Research Institute for Child Health and Development, Tokyo, Japan. [5] Department of Microbiology, College of Life and Health Science, Chubu University, Aichi, Japan. [6] Swiss Institute of Bioinformatics, Basel, Switzerland. *A list of authors and their affiliations appears at the end of the paper. ✉email: janhendrik.niess@unibas.ch

Macrophages are one of the most abundant cell populations in the colonic lamina propria, where they remove apoptotic cell bodies, survey the intestinal content, and ingest and kill microbes that have passed the epithelial barrier[1]. In order to maintain the integrity of the colonic barrier, which is in constant contact with the environment, monocytes continually replenish the tissue-resident macrophage pool[2]. These monocytes originate from blood, enter the gut, and mature into short-lived colonic macrophages locally[3,4], or to long-lived Tim-4 and surface CD4 expressing lamina propria macrophages[5] and to long-lived submucosa and myenteric plexus macrophages, which also could be remnants of embryonic-derived macrophages[6]. This phenomenon is an exception to most other tissues, where macrophages are embryonically derived, self-renewed, and not replaced continuously by monocytes[7–9].

Adoptively transferred monocytes from Cx3cr1-GFP mice, tracked in recipient animals depleted in CD11c[+] cells, give rise to lamina propria CX3CR1[+] macrophages[3,4]. Tracking of monocytes after adoptive transfer later suggested that monocytes mature to lamina propria F4/80[+] CX3CR1[high] MHCII[+] macrophages through different stages of monocyte intermediates following a "monocyte waterfall"-development[10]. Accordingly, monocytes stepwise downregulate the expression of Ly6C[high] and acquire F4/80, CD64, CD11c, and CX3CR1[10]. However, the sequence of molecular events leading to the stepwise differentiation of monocytes into lamina propria macrophages is yet to be understood.

Several lines of evidence have implicated that sensing of nutrients by macrophages assists their development[11]. For example, the influx of leucine in human macrophages or the constitutive activation of the mammalian target of rapamycin complex 1 (mTORC1) in mice promotes the production of pro-inflammatory cytokines[12,13]. It has recently been shown, that the branched-chain amino acid transporter CD98, composed of a heavy (CD98hc) and light (CD98lc) subunit, is highly expressed by monocytes and macrophages, and plays an essential role in the activation and functions of macrophages[14]. We thought that the deletion of CD98hc in CX3CR1[+] intestinal macrophages might allow studying the requirement of branched-chain amino acids for their development and maturation. Because both monocytes and macrophages express CX3CR1[10], we bred CD98hc[flox/flox] with Cx3cr1[CreER]-YFP mice to conditionally delete CD98hc (cKO) in these populations.

The glycoprotein CD98hc, which was termed initially 4F2 and identified as an activation antigen of lymphocytes[15], is an integral membrane protein that contains a single-pass heavy chain (encoded by the genes SLC3A2 for human and Slc3a2 for mouse), which is covalently linked to a multi-pass light chain (LAT1 and LAT2, encoded by the genes SLC7A5/Slc7a5 and SLC7A8/Slc7a8, respectively) via a disulfide bond[16]. It is now clear that the role of CD98hc extends beyond being only an activation marker; for example, it is required for clonal expansion of T cells and B cells[17,18]. Furthermore, CD98hc also binds to β1A and β3 integrins[19], which mediates adhesive signals to the local microenvironment in the bone marrow niche and thereby facilitates the progression of acute myelogenous leukemia[20].

In the context of inflammatory bowel disease, the overexpression of CD98hc in intestinal epithelial cells leads to exacerbated colitis and colitis-associated cancer[21]. On the other hand, the oral treatment of mice with nanoparticles loaded with CD98hc small interfering RNA attenuated the severity of colitis[22]. If the conditional deletion of CD98hc affects the development of CX3CR1[+] intestinal monocytes and macrophages, presumably, this will impact the colitis severity.

In this study, we describe the transcriptional landscape of the "monocyte waterfall"-development to mature colonic macrophages by single-cell RNA sequencing. We show that the conditional deletion of CD98hc in CX3CR1[+] monocytes and macrophages altered their development to mature MHCII[+] macrophages, increased the expression of apoptotic genes and attenuated colitis.

## Results

### CD98hc expression in colonic macrophages and progenitors.

Ex vivo bone marrow cells from unmanipulated C57BL/6 mice were analyzed by flow cytometry to measure CD98hc expression levels in monocyte–macrophage dendritic cell progenitors, in common monocyte progenitors, and in monocytes[23]. The bone marrow cells were separated into Lin[−] CD115[+] CD117[+] CD135[+] Ly6C[−] CD11b[−] monocyte–macrophage dendritic cell progenitors, Lin[−] CD115[+] CD117[+] CD135[−] Ly6C[+] CD11b[−] common monocyte progenitors, and Lin[−] CD115[+] CD117[+] CD135[+] CD11b[+] monocytes, which were further grouped into Ly6C[high], Ly6C[mid], and Ly6C[low] monocytes (Fig. 1a). Most monocyte–macrophage dendritic cell progenitors, common monocyte progenitors and Ly6C[high] monocytes were CD98hc positive. The Ly6C[mid] and Ly6C[low] monocytes displayed a lower proportion of CD98hc-positive cells, and lower CD98hc median fluorescence intensity (Fig. 1b, c). The distribution of markers used for the delineation of respective cell populations in the bone marrow is shown in Supplementary Fig. 1a. Colonic macrophages originate from the extravasation of blood monocytes into the lamina propria, which then passes through the "monocyte waterfall" differentiation phases to become resident gut macrophages[24]. Cells of the colonic lamina propria can be separated into a Lin[−] CD11b[+] CCR2[+] population, which corresponds to extravasated blood monocytes and transit from Ly6C[high] MHCII[−], via Ly6C[mid] MHCII[+], to Ly6C[low] MHCII[+] subpopulations, and Lin[−] CD11b[+] CD64[+] population. The CD11b[+] CD64[+] population is composed of two subpopulations, MHCII[−] and MHCII[+] (Fig. 1d, respective staining control Supplementary Fig. 1b). The distribution of markers used for the delineation of respective cell populations in the colonic lamina propria is shown in Supplementary Fig. 1c.

Scanning electron microscopy revealed that the CD64[+] MHCII[−] and MHCII[+] cells are larger in size compared to Ly6C[high], Ly6C[mid], and Ly6C[low] monocytes (Fig. 1e). Altogether, flow cytometry and scanning electron microscopy images indicated that MHCII[−] and MHCII[+] CD64[+] cell populations are intestinal macrophages characterized by their marker expression as well as the cell size[25]. The distinct subpopulations showed a high CD98hc expression level and, although the CD64[+] MHCII[−] subpopulation has fewer CD98hc[+] cells, there were no notable differences in their CD98hc mean fluorescence intensity (Fig. 1f, g).

Immunofluorescence staining confirmed that CX3CR1[+] monocytes and macrophages, as well as epithelial cells, express CD98hc (Supplementary Fig. 2a). Bone marrow-derived macrophages were generated in the presence of M-CSF to determine the expression level of CD98hc in vitro. The differentiated bone marrow-derived macrophages were stimulated with LPS + IFN-γ or with IL-4 + IL-13 for "M1" and "M2" polarization, respectively. Flow cytometric analysis showed that LPS + IFN-γ, as well as the IL-4 + IL-13 stimulation of bone marrow-derived macrophages, did not modulate CD98hc expression (Supplementary Fig. 2b, c). Since a fraction of tissue-resident macrophages may originate from the embryonic yolk sac, the CD98hc expression was determined in the macrophages of the embryonic yolk sac (E8.5), liver myeloid cells, and embryonic-derived resident Langerhans cells to compare the expression levels during maturation of the animals. The macrophages of the yolk sac showed a lower CD98hc expression than the tissue-resident liver myeloid cells and Langerhans cells that were >95% positive for

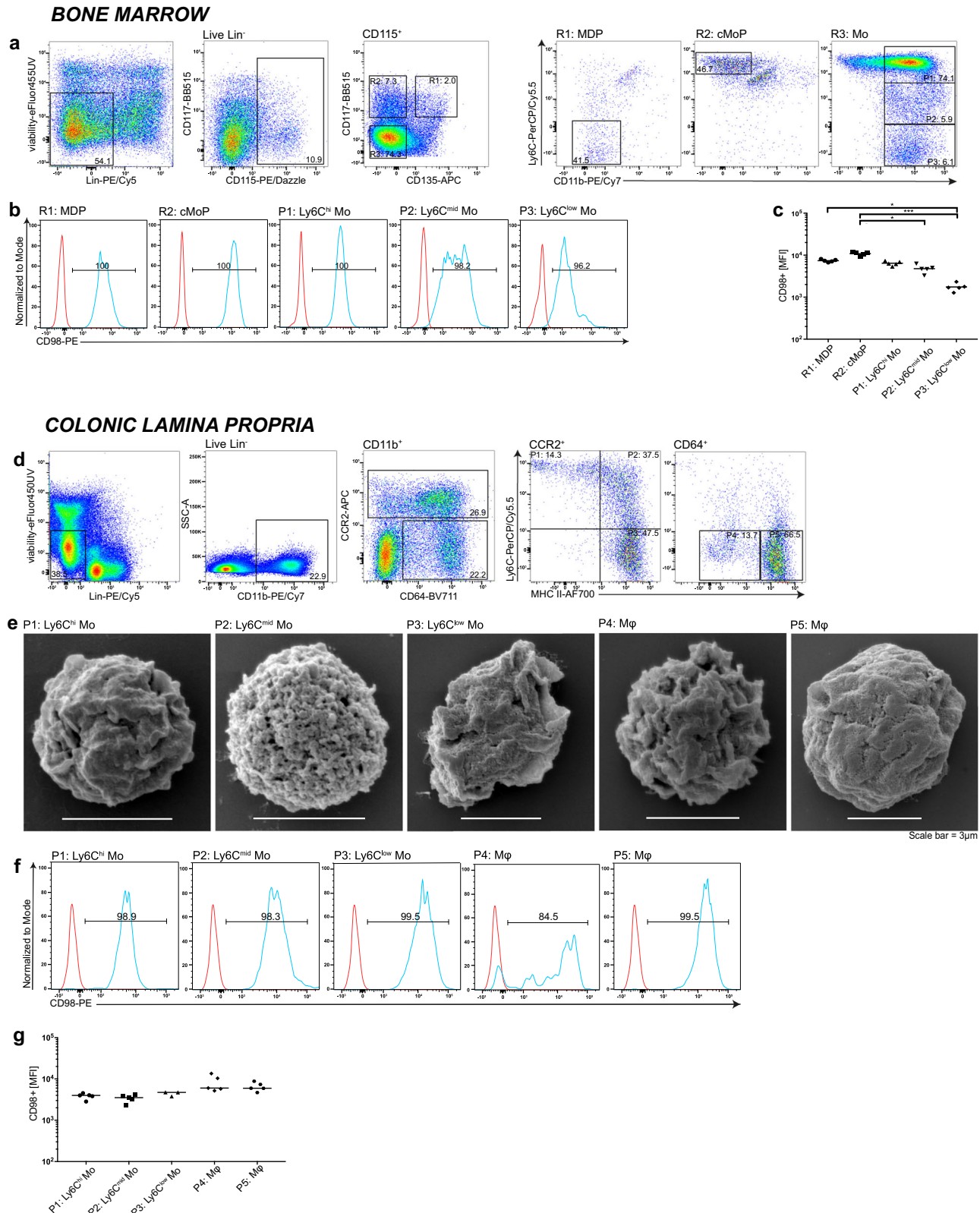

**BONE MARROW**

**COLONIC LAMINA PROPRIA**

Scale bar = 3μm

CD98hc (Supplementary Fig. 2d–f). Taken together, our results show that mononuclear phagocytes and their progenitors express CD98hc in steady-state as well as in inflamed conditions.

**CD98hc deletion in colonic macrophages**. We next aimed to establish a mouse model allowing the deletion of CD98hc specific

in monocytes and macrophages to investigate the effect of CD98hc on their development. Colonic macrophages and their progenitors express the chemokine receptor CX3CR1[26], whose ligand fractalkine/CX3CL1 is expressed by intestinal epithelial cells[26,27]. We generated CD98hc$^{\Delta CX3CR1}$ mice by breeding Cx3cr1$^{CreER}$-YFP mice with CD98hc$^{flox/flox}$ mice for the tamoxifen-inducible conditional deletion of CD98hc[14]. After

**Fig. 1 Monocytes, macrophages, and their progenitors express CD98hc.** Bone marrow cells were isolated from C57Bl/6 wild-type (WT) mice. Monocyte–macrophage dendritic cell progenitors, (MDP), common monocyte progenitors (cMoP) and monocytes (Mo) were analyzed for CD98hc expression. **a** After gating on viable, CD115$^+$ and lineage-negative cells, MDPs were identified as, CD117$^+$, CD135$^+$, Ly6C$^-$, and CD11b$^-$ cells. cMoPs were defined as CD117$^+$, CD135$^-$, Ly6C$^+$, and CD11b$^-$ cells, and monocytes characterized as CD117$^-$, CD135$^-$, and CD11b$^+$ cells with Ly6C$^{high}$, Ly6C$^{mid}$, and Ly6C$^{low}$ expression. **b** Expression and **c** median fluorescence intensity (MFI) of the glycoprotein CD98hc by indicated monocytes and their progenitors. ($n = 5$ independent animals). **d** In the colonic lamina propria, after gating on viable and lineage-negative population, CD11b$^+$ cells were used for further classification. CCR2/CD64 dot plots were obtained by gating on CD11b$^+$ cells to discriminate CCR2$^+$/CD64$^-$ and CCR2$^+$/CD64$^+$ monocytes from CD64$^+$/CCR2$^-$ macrophages (Mφ), which were further distinguished by Ly6C and MHCII staining. **e** Representative scanning electron microscopy images of colonic monocytes and macrophages. **f** Expression and **g** MFI of CD98hc in distinct populations FMO controls are indicated by red histograms, blue histograms indicate CD98hc stained cells. Numbers in histogram plots indicated the percentage of CD98hc$^+$ cells (**b**, **f**). Each dot represents one independent animal; the mean is indicated. The data were analyzed by Kruskal–Wallis test followed by Dunn's correction; *$p < 0.05$, ***$p < 0.001$ (**c**, **g**). Experiments were performed thrice with three to five biological replicates in each group.

breeding the CD98hc$^{flox/flox}$ with Cx3cr1$^{CreER}$-YFP mice (strain thereafter named CD98hc$^{ΔCX3CR1}$ mice), the tamoxifen-induced Cre-mediated recombination led to the excision of exon 3 of *Cd98hc* (Supplementary Fig. 3a, b). Of note, the injection of tamoxifen into pregnant CD98hc$^{ΔCX3CR1}$ mice was intrauterine lethal to the offspring (data not shown). We administered tamoxifen in its carrier corn oil every 24 h for 5 consecutive days and determined the CD98hc expression in colonic monocytes and tissue macrophages on day 2, 7, 14, and 21 after first tamoxifen injection. We observed a decreased percentage of CD98hc$^+$ monocytes and macrophages subpopulations of the colonic lamina propria already after 2 days, and the lowest percentage was observed at day 7 (Fig. 2a, b). On day 14, the percentage of CD98hc$^+$ Ly6C$^{high}$ and Ly6C$^{mid}$ monocytes had returned to normal, whereas for CD98hc$^+$ Ly6C$^{low}$ monocytes, this took 21 days. By contrast, the CD98hc expression of MHCII$^-$ and MHCII$^+$ macrophages in the colonic lamina propria did not fully recover within 21 days. Of note, the distribution of CD98hc intensity in macrophages on day 14 was bimodal, suggesting that silenced macrophages are replaced with newly recruited CD98hc$^+$ cells (Fig. 2b). It has been reported that macrophages, monocytes, T cell subsets, B cells, NK cells, dendritic cells, and platelets express the fractalkine receptor CX3CR1[28]. Therefore, we investigated the effect of deletion on the expression of CD98hc in different intestinal immune cell populations under inflamed conditions to exclude the possibility that tamoxifen injection into CD98hc$^{ΔCX3CR1}$ mice also deletes CD98hc in other immune cell populations. Colonic macrophages but not T and B cells showed a substantial reduction in CD98hc expression after tamoxifen treatment (Supplementary Fig. 3c, d). As CD98hc binds to integrin β1 we verified that conditional deletion of CD98hc did not affect integrin β1 expression across the monocyte and macrophage subpopulations in the colonic lamina propria of mice with colitis (Supplementary Fig. 3e).

We observed that tamoxifen treatment also led to the depletion of CD98hc in CD11c$^-$ and CD11c$^+$ liver myeloid cells (Fig. 2c, e). Flow cytometry confirmed that the major part of liver myeloid cells expresses CX3CR1/YFP along with the low expression of zinc finger E-box binding homeobox 2 (Zeb2) in contrast to Kupffer cells that have high Zeb2 and lack CX3CR1/YFP[29,30] (Fig. 2f, g). Yolk-sac-derived myeloid progenitors, fetal-liver-derived monocytes, and hematopoietic stem cell-derived myeloid precursors contribute to the development of tissue-resident macrophages and epidermal Langerhans cells[7]. We tested whether tamoxifen injection affects the CD98hc expression in embryonic-derived epidermal Langerhans cells, brain microglia, and bone marrow-derived cardiac macrophages. Tamoxifen injection did not substantially affect the CD98hc expression in embryonic-derived epidermal Langerhans cells (Fig. 2d, e) and brain microglia (Supplementary Fig. 4a, b), but reduced CD98hc expression in bone marrow-derived cardiac macrophages

(Supplementary Fig. 4a, b). Overall, these results indicate that we have established a mouse model (CD98hc$^{ΔCX3CR1}$), in which tamoxifen injection successfully deletes the expression of CD98hc in a relatively selective manner in most CX3CR1/YFP macrophage populations.

**CD98hc expression in inflammatory bowel disease.** To pursue these results further, we sought to test the clinical relevance of CD98 expression in the colonic mucosa in patients with inflammatory bowel disease. Biopsies from healthy individuals, from patients with quiescent and active ulcerative colitis and patients with quiescent and active Crohn's disease were taken from non-inflamed and inflamed regions to determine the expression level of CD98 heavy chain (*CD98hc/SLC3A2*) and CD98 light chain (*CD98lc/SLC7A5*). The RT-qPCR analysis revealed profoundly higher *CD98hc* and *CD98lc* expression in ulcerative colitis and Crohn's disease with both quiescent and active disease compared with healthy individuals (Fig. 3a). Immunofluorescence staining confirmed the increased expression of CD98hc by intestinal epithelial cells as well as lamina propria cells in patients with ulcerative colitis and Crohn's disease compared with healthy individuals (Fig. 3b, c). Taken together, these data show that CD98hc and CD98lc are expressed in the human colonic lamina propria, and display high expression in patients with quiescent and active inflammatory bowel disease.

**Attenuated colitis after CD98hc deletion in macrophages.** We next determined whether the conditional deletion of CD98hc in intestinal monocyte and macrophage populations influenced the development of Dextran Sodium Sulfate-induced colitis. We treated both CD98hc$^{ΔCX3CR1}$ and CD98hc$^{flox/flox}$ animals either with tamoxifen or the carrier corn oil as a control for 2 days before dextran sodium sulfate administration. Immunofluorescence staining and flow cytometry verified the successful deletion of CD98hc by tamoxifen in intestinal monocyte and macrophage populations of these mice (Fig. 4a, b). Tamoxifen-treated CD98hc$^{ΔCX3CR1}$ mice showed a substantial reduction in cardinal markers of colitis, such as body weight loss (Fig. 4c), disease activity index (Fig. 4d), histological signs of colitis (Fig. 4e, h), and colon shortening (Fig. 4g, f) compared with the control groups. Hematoxylin staining (Fig. 4h) and colonoscopy (Fig. 4i) confirmed the decreased severity of colitis in tamoxifen-treated CD98hc$^{ΔCX3CR1}$ mice, compared with corn oil-treated mice.

We next fed animals with chow enriched with 5% more L-leucine and 5% more L-isoleucine compared to regular chow, and induced dextran sodium sulfate colitis in these animals. We compared the colitis severity of wild-type C57Bl6 (WT), corn oil or tamoxifen-treated CD98hc$^{flox/flox}$, or tamoxifen-treated CD98hc$^{ΔCX3CR1}$ animals receiving either regular chow or a high amino acid diet. We did not observe notable differences in disease

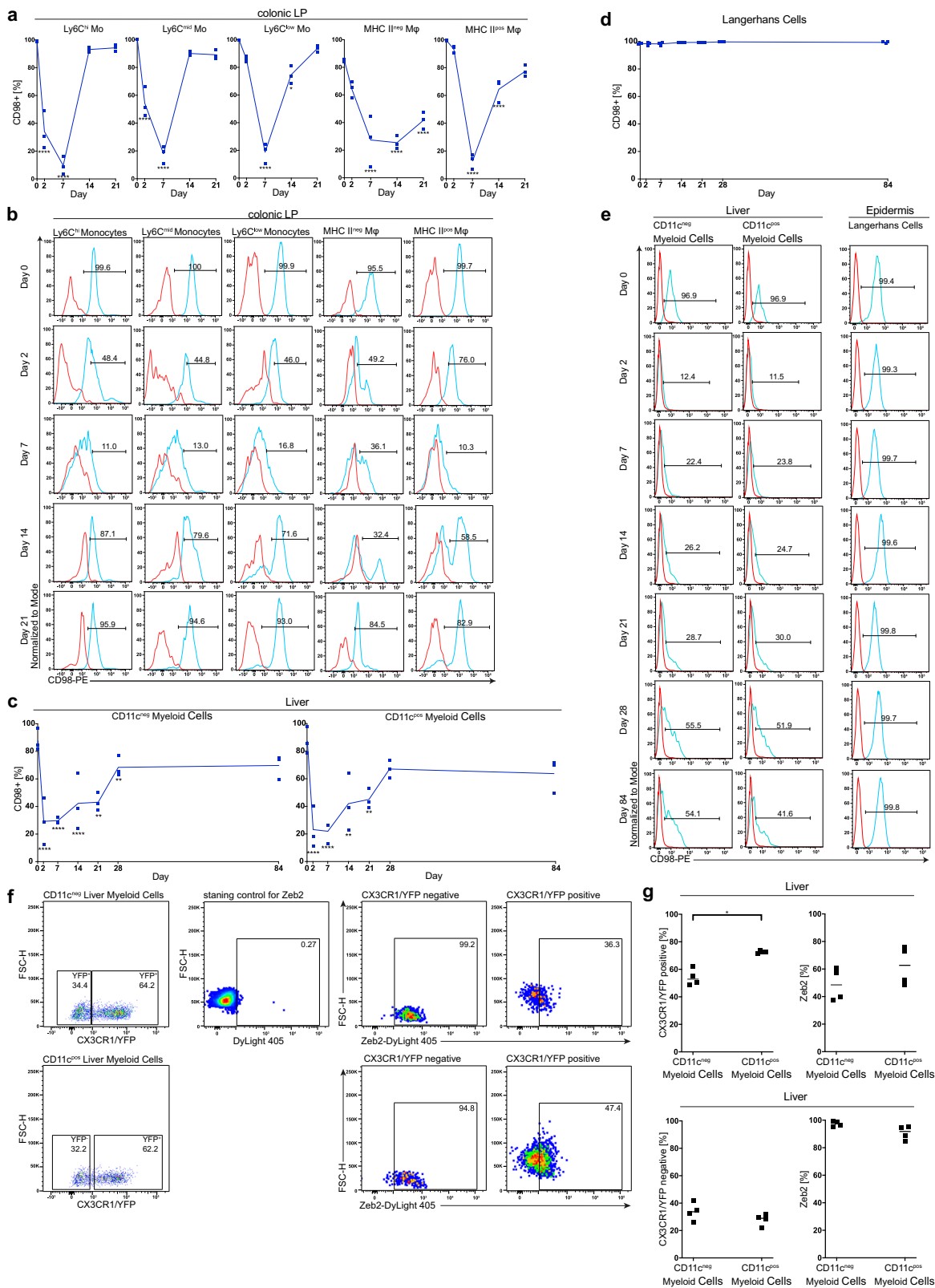

activity scores, colon length, histological scores between groups, except for tamoxifen-treated CD98hc$^{\Delta CX3CR1}$ animals with a high amino acid diet, which exhibited higher disease activity scores (Supplementary Fig. 5a–c). Of note, immunofluorescence staining suggested that conditional deletion of CD98hc in macrophages led to reduced macrophage numbers in the colonic lamina propria (Supplementary Fig. 5d, e). To investigate possible

molecular pathways downstream of CD98hc, we first measured the phosphorylation of the ribosomal protein S6 kinase beta-1 (p70S6K), which is a downstream target of mTOR complex activated by amino acids. The conditional deletion of CD98hc in mice with colitis did not influence the phosphorylation of p70S6K in macrophages (Supplementary Fig. 6a). Further, the deprivation of amino acids by culturing bone marrow-derived macrophages

**Fig. 2 Tamoxifen injection into CD98hc$^{\Delta CX3CR1}$ animals leads to the excision of CD98hc in monocytes and macrophages.** Following intraperitoneal tamoxifen injection, monocytes, and macrophages were isolated from the colonic lamina propria (cLP) of CD98hc$^{\Delta CX3CR1}$ animals at indicated time points and analyzed for CD98hc expression by flow cytometry. **a** Percentage of CD98hc$^+$ monocytes and macrophages ($n = 3$). **b** Histogram plots showing CD98hc staining intensity in cLP monocytes and macrophages. **c** Mean (±SD) percentage of CD98hc$^+$ CD11c$^{neg}$, and CD11c$^{pos}$ liver myeloid cells and (**d**) Langerhans cells ($n = 3$). **e** Histogram plots of CD11c$^{neg}$ and CD11c$^{pos}$ liver myeloid cells and Langerhans cells isolated from liver and epidermis, respectively. Red histograms display FMO controls, blue histograms CD98hc$^+$ cells; Numbers in histograms show the percentage of CD98hc$^+$ cells (**b**, **e**). The data are shown as the mean (± SD), and the results were analyzed by two-way ANOVA followed by Sidak's correction; *$p < 0.05$, **$p < 0.01$, ***$p < 0.001$, ****$p < 0.0001$ (**a–d**). **f** After gating on CX3CR1/YFP$^+$ or CX3CR1/YFP$^-$ CD11c$^+$ or CD11c$^-$ liver myeloid cells, Zeb2 expression by CD11c$^+$ or CD11c$^-$ liver myeloid cells were analyzed. Numbers in dot plots indicate the percentage of positive cells. **g** Percentage of CX3CR1/YFP$^+$ or CX3CR1/YFP$^-$, and percentage of Zeb2$^+$ CD11c$^+$ or CD11c$^-$ liver myeloid cells. Each dot represents one animal; the mean is indicated. The data were analyzed by Mann–Whitney U test; *$p < 0.05$. Experiments were performed once with three biological replicates for CD98hc silencing kinetics, and once with four biological replicates for Zeb2 expression.

in nonessential amino acid medium did not appreciably influence the phosphorylation of the p70S6K, its substrate S6 ribosomal protein (S6) and production of *Tnf, iNos, Il6, Mcp1, Il1β, Kc*, and *Il1α*. The supplementation of leucine to bone marrow-derived macrophages cultured in nonessential amino acid medium did not affect the mTORC1 pathway but promoted *Il1α* expression in vitro (Supplementary Fig. 6b–d). Altogether, these results indicate that the loss of CD98hc in CX3CR1$^+$ macrophages attenuated dextran sodium sulfate-induced colitis in mice.

**Developmental trajectory of monocytes to colonic macrophages.** To further characterize the effects of conditional deletion of CD98hc in monocytes and macrophages, we sorted CD11b$^+$ cells expressing either CCR2 or CD64 isolated from the colonic lamina propria of CD98hc$^{\Delta CX3CR1}$ female littermates after treatment with tamoxifen or corn oil as a control for 5 consecutive days. We performed single-cell RNA sequencing (scRNA-seq) with the 10× Genomics technology using four mice per condition 7 days after the first tamoxifen injection (Fig. 5a). All samples were processed on the same day on the eight wells of the same cartridge to exclude the possibility of confounding batch effects. After quality filtering, the resulting data set included expression values for 11,947 genes in 3213 cells, ranging from 83 to 724 cells per sample, for a total of 1863 control cells and 1350 CD98hc cKO cells. An average of 5452 UMIs (ranging from 631 to 27,487) was sequenced per cell. An average of 1645 genes (ranging from 399 to and max. 4541) were detected per cell (Supplementary Data 1 (10× genomics web summaries of colon 1 to 8) and Supplementary Table 1). Overall, we observed a sufficient overlap of cells from different biological replicates in each condition (Supplementary Fig. 7a).

Unsupervised hierarchical clustering of the cells followed by dynamic dendrogram cutting yielded nine clusters (Fig. 5b), all including cells from control and CD98hc cKO samples. We annotated the individual cells by comparing them to the immunological genome project (ImmGen) reference data set[31]. This analysis revealed that clusters 1 and 4 were mostly composed of monocytes and macrophages, cluster 2 of monocytes, and cluster 3 and 6 of macrophages. Other clusters likely gathered contaminants, with cluster 5 mostly composed of DCs, cluster 7 of monocytes, macrophages, and dendritic cells, cluster 8 of innate lymphoid cells, T cells, and B cells and cluster 9 of fibroblasts and stromal cells (Fig. 5c; Supplementary Data 2 and Supplementary Table 2). The expression levels of known markers and cluster-specific genes further confirmed the cell-type identity of clusters 1, 2, 3, 4, and 6 (Fig. 5d; Supplementary Figs. 7b and 8), with notably, *Ccr2, Cd14*, and *Ly6c2* expressed by cluster 1, 2, and to a lower extent by cluster 4, confirming that these cells are monocytes. *Cd63, Cd72, Cd74, Cd81*, EYFP (which is in CD98hc$^{\Delta CX3CR1}$ mice under control of the *Cx3cr1* promoter), *Cx3cr1*, and *Adgre1* (EMR1, F4/80) were expressed by clusters 3 and 6 confirming that these cells are macrophages (Supplementary Fig. 7b). The long-

lived lamina propria macrophage marker *Cd4* was expressed by few cells of cluster 3 and 6 (macrophages) and of cluster 8 (innate lymphoid cells and T cells), and few cells of cluster 3 and 6 (macrophages), and of cluster 5 (dendritic cells) express *Timd4* (Fig. 5d; Supplementary Fig. 8). Relevant patterns observed in the scRNA-seq data were verified by flow cytometry. MHCII$^-$ and MHCII$^+$ macrophages have higher CD81 expression compared with Ly6C$^{high}$ and Ly6C$^{mid}$ monocytes. Although monocytes downregulated *Cd14* and upregulated *Cd72* during their development into macrophages, at the protein level monocytes and macrophages expressed CD14 and CD72 at similar levels (Supplementary Fig. 9a, b).

The observation of cells on principal component analysis (Fig. 5e; Supplementary Data 3 (Plotly Data Visualization)), as well as the patterns of expression of cluster-specific genes (Supplementary Fig. 7b) and a FlowSOM analysis (Fig. 5f) suggested a differentiation trajectory from clusters 1 and 2 either to clusters 3 or to cluster 6. The comparison of relative proportions of control and cKO cells across clusters and across the principal components 1 and 2 space, indicated the most substantial enrichment of cKO cells in cluster 2, at the beginning of the differentiation trajectory, and the most robust enrichment of control cells in cluster 3, at the end of the differentiation trajectory (Fig. 5g; Supplementary Fig. 9c). Overall, these observations suggest that the deletion of CD98hc in monocytes and macrophages resulted in an altered "monocyte waterfall"-development into mature macrophages in the colonic lamina propria in tamoxifen-treated CD98hc$^{\Delta CX3CR1}$ mice, which is also apparent when the relative proportions of control and cKO cells are projected onto the nodes of the FlowSOM tree (Fig. 5h).

**Increased apoptotic signatures upon CD98hc deletion.** To further gain insights into molecular mechanisms involved in the altered development in CD98hc-deficient macrophages, we looked within each cluster along the developmental trajectory for genes differentially expressed between CD98hc$^{\Delta CX3CR1}$ mice treated with tamoxifen or corn oil as control, using a "pseudo-bulk" approach[32]. The "pseudo-bulk" samples used for this analysis were first controlled on a principal component analysis, where the two first principal components separated the samples according to their differentiation stage, and principal components 3 and 4 separated the control and cKO samples (Fig. 6a). Supplementary Data 4 list the genes in each individual cluster that were substantially up- or downregulated between cKO and control samples. A gene set enrichment analysis on the results of the differential expression analysis showed a general increase in inflammation in cKO samples. More specifically, apoptosis-associated genes were upregulated in cKO samples of cluster 2 monocytes (Fig. 6b). A selection of known genes involved in apoptosis is displayed in Fig. 6c, d. The expression levels of these genes were overall slightly higher in monocytes of clusters 1 and 2 compared with macrophages of clusters 3, 4, and 6 (Fig. 6b–d).

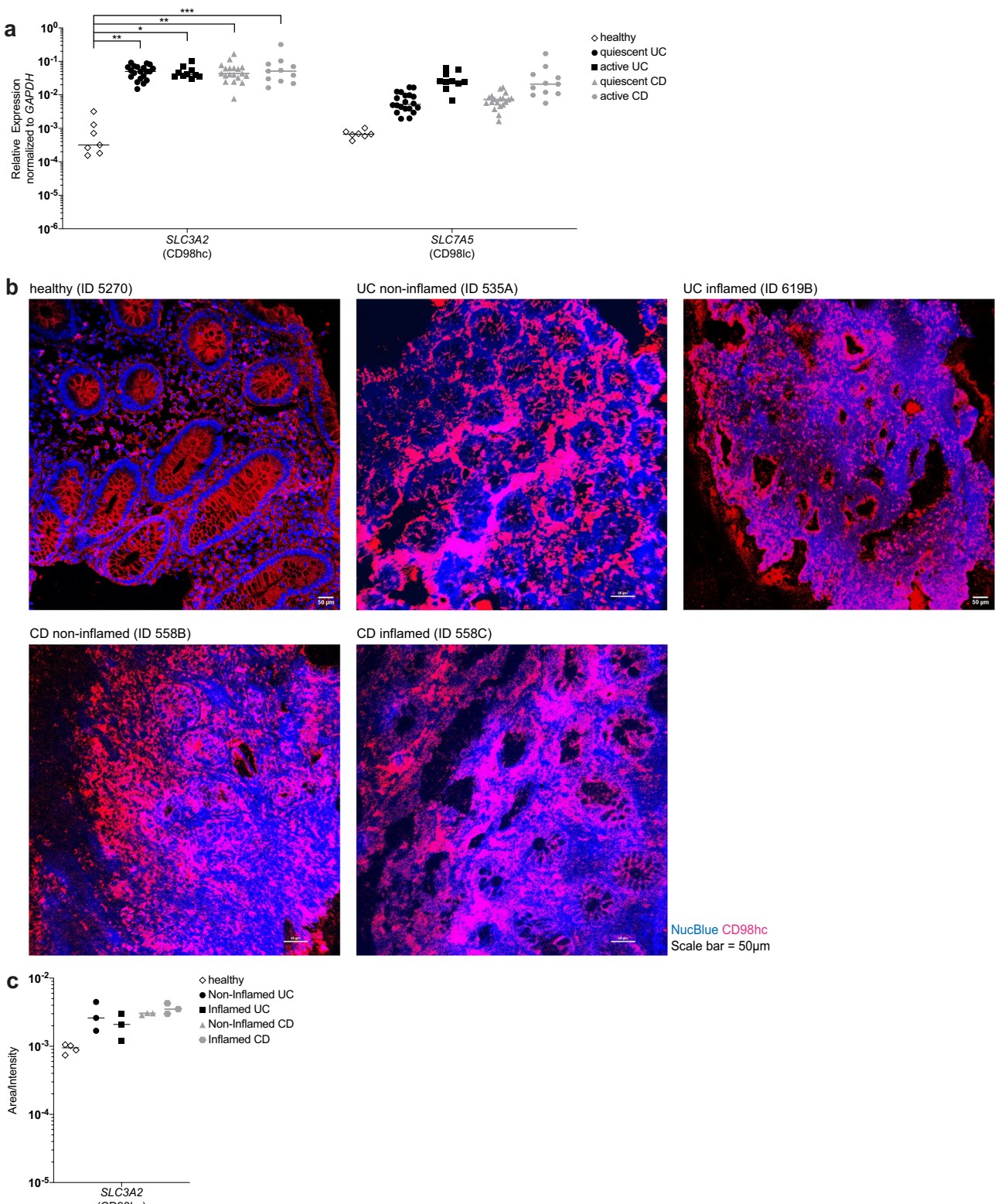

**Fig. 3 Inflammatory bowel disease patients express CD98.** The Swiss IBD cohort study provided colonic or ileal biopsies from Crohn's disease (CD) or ulcerative colitis (UC) patients which were in remission (quiescent) or with active disease. Healthy patients were recruited at the University Hospital Basel. **a** *CD98hc/SLC3A2* and *CD98lc/SLC7A5* expression was determined by qRT-PCR. **b** Cryosections of inflamed and non-inflamed regions of the same CD or UC patient (patient identification numbers in brackets) were stained for CD98hc and counterstained with NucBlue. Immunofluorescence was carried out with biopsies of five healthy patients, and four UC and four CD patients. **c** CD98hc fluorescence intensity of staining of biopsies from CD and UC patients. In the figures of the panels (**a**) and (**c**), the mean is indicated with each dot representing one patient. The data were analyzed by Mann–Whitney *U* test; *$p \leq 0.05$, **$p \leq 0.01$, ***$p \leq 0.001$.

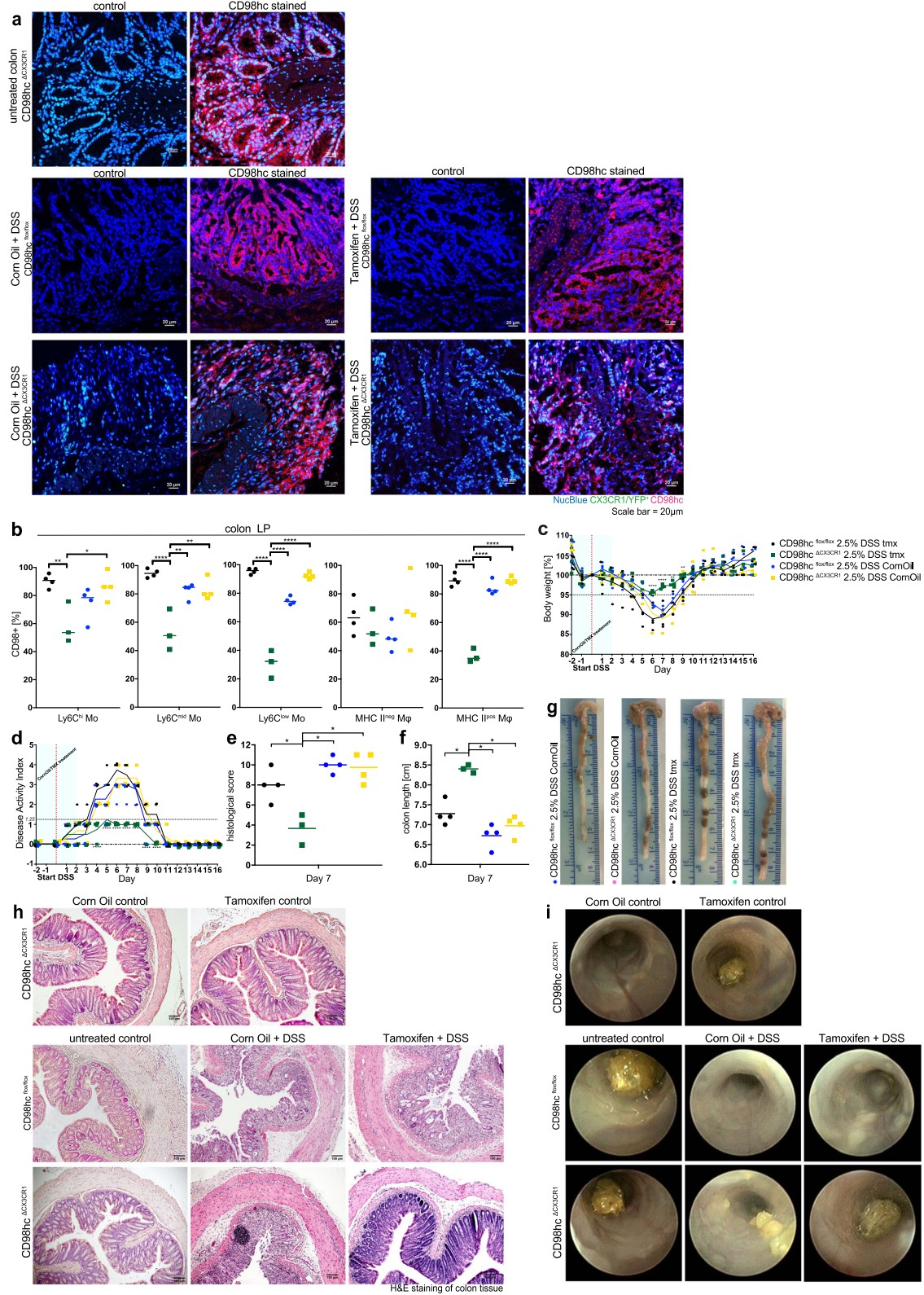

We confirmed increased apoptosis at each stage of the "monocyte waterfall"-development after conditional deletion of CD98hc in an independent experiment in mice using the Dextran Sodium Sulfate colitis model by flow cytometry in vivo (Supplementary Fig. 10a, b). Furthermore, the in vitro conditional deletion of CD98hc by treating M-CSF differentiated bone marrow-derived macrophages obtained from CD98hc^ΔCX3CR1

mice with tamoxifen resulted in cell death (Supplementary Fig. 11a).

**Reduced colonic macrophages after CD98hc deletion**. We next aimed at confirming the altered "monocyte waterfall"-development to mature macrophages in the colonic lamina propria upon

**Fig. 4 Conditional deletion of CD98hc in monocytes and macrophages leads to attenuated colitis. a** CD98hc immunofluorescence of CD98hc$^{\Delta CX3CR1}$ mice, CD98hc$^{flox/flox}$, and CD98hc$^{\Delta CX3CR1}$ with DSS colitis treated either with corn oil or tamoxifen. **b** Percentage of CD98hc$^+$ monocytes and macrophages of indicated groups 7 days after start of DSS administration. Colitis was induced by adding 2.5% dextran sodium sulfate (DSS) to CD98hc$^{flox/flox}$ and CD98hc$^{\Delta CX3CR1}$ mice which were treated either with corn oil or tamoxifen 2 days before DSS administration. Each dot represents one animal. Data were analyzed by two-way ANOVA followed by Sidak's correction; *$p < 0.05$, **$p < 0.01$, ***$p < 0.001$. **c** The mean percentage body weight change (± SD) and **d** disease activity index are shown. The data were analyzed by two-way ANOVA followed by Sidak's correction; ****$p < 0.0001$. **e** Histological scores were assessed in a blinded fashion by two independent investigators. The mean histological score was determined for each animal after H&E staining of colonic tissues, presented as individual dot and analyzed with a Mann–Whitney U test; *$p < 0.05$. **f** The colon length was determined at day 7 after start of DSS administration, colon length is shown for each individual animal, the mean indicated and analyzed with a Mann–Whitney U test; *$p < 0.05$. **g** A representative image of the colon from each group is shown. **h** Hematoxylin staining of colonic tissues, and **i** endoscopic images from indicated groups. Experiments were performed four times with three to four biological replicates in each group.

conditional deletion of CD98hc. Daily injection of tamoxifen throughout 28 days deleted CD98hc constantly in monocytes and macrophages (Fig. 7a). We then determined the number of monocyte and macrophage populations in tamoxifen- and corn oil-treated CD98hc$^{\Delta CX3CR1}$ animals at indicated time points. Ly6C$^{mid}$ and Ly6C$^{low}$ monocytes and MHCII$^+$ macrophages showed also a substantial reduction of CD98hc expression (Fig. 7a) and additionally a prominent decrease in the cell number (by day 2 for Ly6C$^{mid}$ and by day 7 for Ly6C$^{low}$ monocytes and MHCII$^+$ macrophages) which was less prominent for Ly6C$^{hi}$ monocytes and MHCII$^-$ macrophages (Fig. 7b). After that, we calculated the ratio between the number of cells in tamoxifen-treated (cKO) and corn oil-treated (WT) CD98hc$^{\Delta CX3CR1}$ littermates in monocyte and macrophage at the indicated time points during tamoxifen or corn oil injections. This analysis indicated that CD98hc deletion reduced Ly6C$^{mid}$ and Ly6C$^{low}$ monocytes and MHCII$^+$ macrophages in the colonic lamina propria, which was less prominent for Ly6C$^{hi}$ monocytes and MHCII$^-$ macrophages (Fig. 7c). Flow cytometric analysis also indicated a relative decrease in the percentage of MHCII$^+$ macrophages in tamoxifen-treated CD98hc$^{\Delta CX3CR1}$ littermates (Fig. 7d).

To investigate whether the absence of CD98hc in cKO animals changes the development and maturation of macrophages, we co-transferred CD11b$^+$ bone marrow monocytes from B6 Ly5.1 (CD45.1$^+$) and CD98hc$^{\Delta CX3CR1}$ (CD45.2 YFP$^+$) animals into CCR2$^{-/-}$ recipients (Fig. 8a). Flow cytometry showed reduced relative number of CD45.2 YFP$^+$ MHCII$^+$ macrophages compared with CD45.1$^+$ MHCII$^+$ macrophages in tamoxifen-treated CCR2$^{-/-}$ recipients versus corn oil-treated animals (Fig. 8b). After normalization of the input ratios, the depletion of CD45.2 YFP$^+$ relative to CD45.1$^+$ macrophages was even more evident, suggesting a change in macrophage development following the deletion of CD98hc (Fig. 8c). Together, our data indicated that conditional deletion of CD98hc leads to the alteration of the "monocyte waterfall"-development in the colonic lamina propria, in particular, diminishing the numbers of mature MHCII$^+$ macrophages.

## Discussion

It has been proposed that in the colonic lamina propria monocytes develop through the "monocyte waterfall" intermediates into mature macrophages[2,8,33]. In this study, we used scRNA-seq to define the development of colonic macrophages on the transcriptomic level and conditional mouse lines to study the relevance of the branched-chain amino acid transporter CD98hc in the development of gut macrophages. We found that monocytes entering the colonic lamina propria undergo increased apoptosis during their development, resulting in reduced numbers of MHCII$^+$ macrophages after deletion of CD98hc in macrophages and their progenitors.

Our results indicate a stepwise development of colonic macrophages from monocytes by flow cytometry and single-cell transcriptomics. The discrepancy between the single-cell transcriptomic and protein level in the expression of some markers might be, in part, explained by differential transcriptional regulation during the maturation of macrophages from monocytes[34,35]. In line with our single-cell transcriptomic data, previous studies have suggested that monocytes give rise to macrophages in the lamina propria[10]. However, the underlying molecular mechanism has not been illustrated in detail. Our scRNA-seq data set, together with conditional mouse lines, indicated that the amino acid transporter CD98hc is in part required for the development of MHCII$^+$ macrophages as the numbers of MHCII$^+$ macrophages are reduced in the colonic lamina propria of cKO animals. One possibility is that branched-chain amino acids derived from nutritional cues, the host, or the microbiota, facilitate the differentiation of monocytes into macrophages[12,13], whose adaptation to the gut environment depends on TGFβR signaling[24,36], and whose numbers are reduced in germ-free animals[2,37–39]. At this stage, we cannot explain the deeper molecular mechanisms why the conditional deletion of CD98hc leads to increased apoptosis in colonic macrophage progenitors. Possibly, different metabolic activities of macrophages and their progenitors after the deletion of CD98hc might explain this phenomenon.

The conditional deletion of CD98hc in macrophages attenuated colitis. In line, overexpression of CD98hc in intestinal epithelial cells leads to more severe colitis and colitis-associated cancer[21], and the treatment of animals with nanoparticles carrying siRNA targeting CD98hc reduces colitis severity[22]. Several studies have connected macrophages with inflammatory bowel disease as multiple inflammatory bowel disease risk genes have essential functions for macrophages[40,41]. Recently, the importance of macrophages for the maintenance of the tissue integrity in the colon has been demonstrated by genetic deletion of tolerogenic signals in mice, which resulted in spontaneous colitis[42–44]. Moreover, the infiltration of macrophages in the inflamed lamina propria of mice or inflammatory bowel disease patients with flares depends on the chemokine receptor CCR2[10,45], suggesting that macrophages can also drive colitis by secreting pro-inflammatory cytokines, such as IL-6, IL-1β, and TNF[10,46]. In line with our results, where macrophage numbers are reduced after conditional deletion of CD98hc, CCR2$^{-/-}$ mice have an attenuated colitis[47].

Furthermore, CD98hc regulates integrin signaling by binding β1A and β3 integrins to its cytoplasmic tail and mediates the uptake of branched-chain amino acids[16]. Although the expression of the integrin β1 was not altered after the conditional deletion of CD98hc in macrophages, we did not investigate in detail, if the alteration of macrophage development is due to impaired integrin signaling and/or impaired uptake of branched-chain amino acids. In line, previous work showed that the conditional deletion of

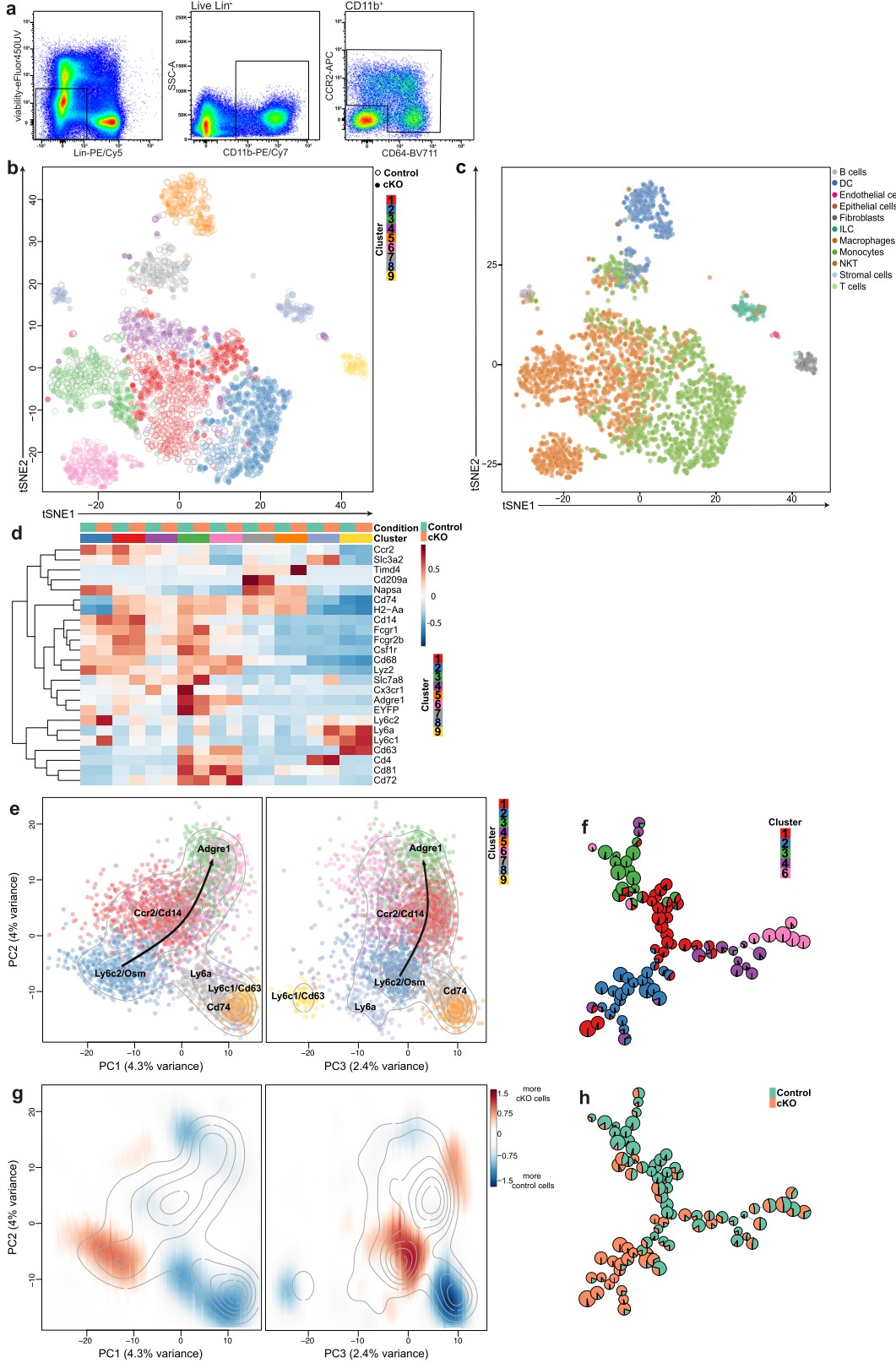

CD98hc in regulatory T cells leads to reduced numbers of regulatory T cells with impaired suppressive function in the gut. This phenomenon was due to reduced uptake of branched-chain amino acids by regulatory T cells and not due to impaired integrin β1-mediated adhesion of regulatory T cells to the local tissue environment of the gut[48]. Hence, the nutritional sensing of

immune cells could modify their functions in unmanipulated and colitis mice. Feeding mice with 10% higher amino acid chow (5% more L-leucine and 5% more L-isoleucine) compared with standard chow revealed a tendency of increased intestinal inflammation in tamoxifen-treated CD98hc$^{\Delta CX3CR1}$ mice. The elevated intestinal inflammation in cKO mice fed with a high

**Fig. 5 Single-cell RNA sequencing suggests a developmental trajectory from monocytes to macrophages in the colonic lamina propria.** After sorting of viable CCR2$^+$CD64$^-$/CCR2$^+$CD64$^+$ and CCR2$^-$CD64$^+$ cells from CD98hc$^{\Delta CX3CR1}$ mice treated with corn oil or tamoxifen, sorted monocytes, and macrophages were further analyzed by single-cell RNA sequencing (scRNA-seq). scRNA-seq was performed in quadruples. **a** Gating strategy for the isolation of CCR2$^+$CD64$^-$/CCR2$^+$CD64$^+$ and CCR2$^-$CD64$^+$ colonic monocytes and macrophages for scRNA-seq analysis. **b** t-SNE analysis depicts the distribution of the nine different clusters and indicates their relationship in corn oil (vehicle control) or tamoxifen (cKO) treated CD98hc$^{\Delta CX3CR1}$ mice. **c** t-SNE visualization shows the annotation of the scRNA-seq data set by using SingleR comparing our data set to the immunological genome project (ImmGen) reference data set. **d** Genes that are characteristic for monocytes and macrophages were depicted and presented as a heatmap. The heatmap of top cluster-specific genes consists of the union of the top ten genes from each between-clusters pairwise comparison. **e** Principal component analysis of single-cells, based on the 500 most variable genes across all cells. The colors represent cells from the different clusters. Contour lines indicate the density of the cells in the principal component analysis space. **f** FlowSOM analysis after exclusion of the contaminant clusters 5, 7, 8, and 9. **g** The color of the differential 2D density plot represents the log2 ratio of 2D densities of cKO cells over control cells. **h** Pie charts within the FlowSOM tree indicate the relative enrichment of cKO cells over control cells. The experiment was performed once (scRNA-seq) with four biological replicates in each group.

amino acid diet is presumably driven by other intestinal immune cells to possibly compensate for the impaired immunological behavior of CD98hc-deficient monocytes and macrophages. Moreover, in vitro cytokine production by bone marrow-derived macrophages underpins the assumption that other immune cells might drive the intestinal inflammation, as bone marrow-derived macrophages cultured in medium containing essential amino acid or medium only supplemented with L-leucine did not show a transition into a pro-inflammatory bone marrow-derived phenotype. It is likely that due to the decrease of colonic monocytes and macrophages, the higher influx of branched-chain amino acid leads to a stronger compensatory and inflammatory response by other colonic immune cells upon CD98hc deletion.

In this study, we show by single-cell transcriptomics that monocytes develop via intermediates and give rise to mature colonic lamina propria macrophages. The conditional deletion of CD98hc in CX3CR1$^+$ macrophages and their progenitors resulted in attenuated chemically-induced colitis and in an altered macrophage development with reduced numbers MHCII$^+$ macrophages in the colonic lamina propria (Supplementary Fig. 12). Looking into the molecular pathways involved in the maturation of gut macrophages may help to understand macrophage development and thereby can further elucidate their roles in inflammatory bowel disease, host defense mechanisms against pathogens, and support the development of oral vaccines.

## Methods

**Animals.** C57Bl/6, CD98hc$^{flox/flox}$, Cx3cr1-GFP (B6.129P2-Cx3cr1$^{tm1Litt/J}$), Cx3cr1$^{CreER}$-YFP (B6.129P2(Cg)-Cx3cr1$^{tm2.1(cre/ERT2)Litt}$/WganJ), B6.Ly5.1 (B6. SJL-Ptprc$^a$ Pepc$^b$/BoyJ), and CCR2$^{-/-}$ (B6.129S4-Ccr2$^{tm1Ifc}$/J) animals were bred and maintained under specific pathogen-free conditions in the animal facility of the Department of Biomedicine, University of Basel, Switzerland. Dr. Hideki Tsumura, Division of Laboratory Animal Resources, Nation Research Institute for Child Health and Development, Tokyo, Japan, provided cryopreserved CD98hc$^{flox/flox}$ embryos[14]. Embryo transfers were conducted in the Center for transgenic animals, University of Basel, Switzerland, and the CD98hc$^{flox/flox}$ mouse line was established. CD98hc$^{flox/flox}$ and Cx3cr1$^{CreER}$-YFP mice were crossed to obtain CD98hc$^{\Delta CX3CR1}$ mice, in which the tamoxifen-inducible, Cre-mediated recombination will lead to the excision of CD98hc (Slc3a2) in CX3CR1$^+$ cells. Female littermates (6–12 weeks of age) were used for the experiments. The local animal welfare committee (animal protocol #2854_27600 (Canton Basel Stadt)) approved the experiments. All experiments were conducted in accordance with the Swiss Federal and Cantonal regulations.

**Patients and study population.** Biopsies from 31 patients with Crohn's disease and 31 patients with ulcerative colitis in RNAlater® stabilization solution (Invitrogen) were received from the Swiss Inflammatory Bowel Disease Cohort Study (Swiss IBD cohort project 2016-12), and stored at −80 °C. This Swiss national cohort of patients with inflammatory bowel disease was started in 2006[49]. Gastroenterologists recruited patients with a diagnosis of Crohn's disease or ulcerative colitis confirmed by endoscopy, radiology or surgery at least four months before inclusion in private practice, regional hospitals, and tertiary centers participating in the Swiss IBD cohort study. Exclusion criteria were other forms of colitis or ileitis, no permanent residency in Switzerland or when informed consent was not admitted. In patients with active inflammatory bowel disease, colonoscopy was performed to assess the activity of the disease and to rule out complications.

In patients with quiescent inflammatory bowel disease, colonoscopy was performed for surveillance. Biopsies were taken from segments that appeared macroscopically inflamed. Supplementary Table 3 gives detailed patient characteristics. The biopsies used for immunofluorescence were obtained from healthy patients, and the inflamed and non- inflamed regions of inflammatory bowel disease patients from the Basel inflammatory bowel disease cohort. Patient characteristics are given in Supplementary Table 4. Informed consent was obtained from all study subjects and the Ethics Committee for Northwest and Central Switzerland (EKNZ) has approved the study (ethic protocol EKBB 139/13 (PB 2016.02242)).

**Conditional deletion of CD98hc.** Tamoxifen (MP Biomedicals) was dissolved in corn oil (Sigma) at a concentration of 20 mg/ml overnight at 37 °C (shaking) and protected from light. The dissolved tamoxifen was stored at 4 °C (protected from light stable up to 1 month). In total, 75 mg tamoxifen/kg body weight were i.p. injected for 5 consecutive days into CD98hc$^{flox/flox}$ x Cx3cr1$^{CreER}$-YFP mice to activate the Cre-recombinase, which leads to the conditional deletion of CD98hc. Control CD98hc$^{flox/flox}$ x Cx3cr1$^{CreER}$-YFP mice received corn oil without tamoxifen.

**Feeding of mice with an amino acid-enriched diet.** Mice were fed by the mouse and rat chow #3436 (extrudate, Granovit AG), which is the standard chow of the animal facility at the Department of Biomedicine in Basel. Five%/kg more L-leucine and 5%/kg more L-isoleucine was supplemented to the standard chow. Mice were fed with the amino acid-enriched diet 4 weeks in advance before starting the experiments, and this diet was continued during the experiments.

**Dextran sodium sulfate-induced colitis.** Dextran sodium sulfate (1.5 to 2.5%; MW: 36,000–50,000; MP Biomedicals) was added to the drinking water of co-housed, weight-matched female (6–12 weeks of age) for 5 days. Dextran sodium sulfate containing water was sterile-filtered before it was given to the animals. On day 5, dextran sodium sulfate in drinking water was exchanged by regular drinking water to induce recovery from colitis. Mice were monitored daily for clinical signs of colitis.

**Clinical colitis score.** Clinical signs of colitis were observed by using the following scores[50]: rectal bleeding: 0—absent, 1—bleeding; rectal prolapses: 0—nil, clear prolapse—mice euthanized; stool consistency: 0—normal, 1—loose stools, 2—diarrhea; position: 0—normal movement, 1—reluctance to move, 2—hunched position; appearance of the fur: 0—normal appearance, 1—ruffled fur, 2—spiky fur; weight: 0—no loss, 1—body weight loss 0–5%, 2—body weight loss >5–10%, 3—body weight loss >10–15%, 4—body weight loss >15%. Once per day, the blinded investigator observed the animals. If the total score was ≥4, the animals were monitored twice per day. The respective animal was euthanized, when the total score was ≥6, when an individual animal lost >15% body weight, when gross bleeding occurred, or when rectal prolapse was noted.

**Endoscopy.** After anesthetizing the mice by intraperitoneal injection of 200 μl, anesthetic solution containing 1 mg/ml xylazine (Xylazin Streuli ad us. vet., injection solution) and 100 mg/ml ketasol (Ketasol®-100 ad us. vet., injection solution) in sterile PBS, the distal 3 cm of the colon and the rectum were examined with a coloview using the tele pack vet X LED RP100 (Karl Storz).

**Hematoxylin staining and histological colitis score.** Colonic tissue was fixed in 4% paraformaldehyde and embedded in paraffin blocks. Six-micrometer sections were stained with hematoxylin. Histological features of colonic inflammation was scored with a previously published scoring system[51]: extent of destruction of normal mucosal architecture (0: normal; 1: mild; 2: moderate; 3: extensive damage), presence and degree of cellular infiltration (0: normal; 1: mild; 2: moderate; 3: transmural infiltration), extent of muscle thickening (0: normal; 1: mild; 2: moderate; 3: extensive thickening), presence or absence of crypt abscesses (0: absent; 1:

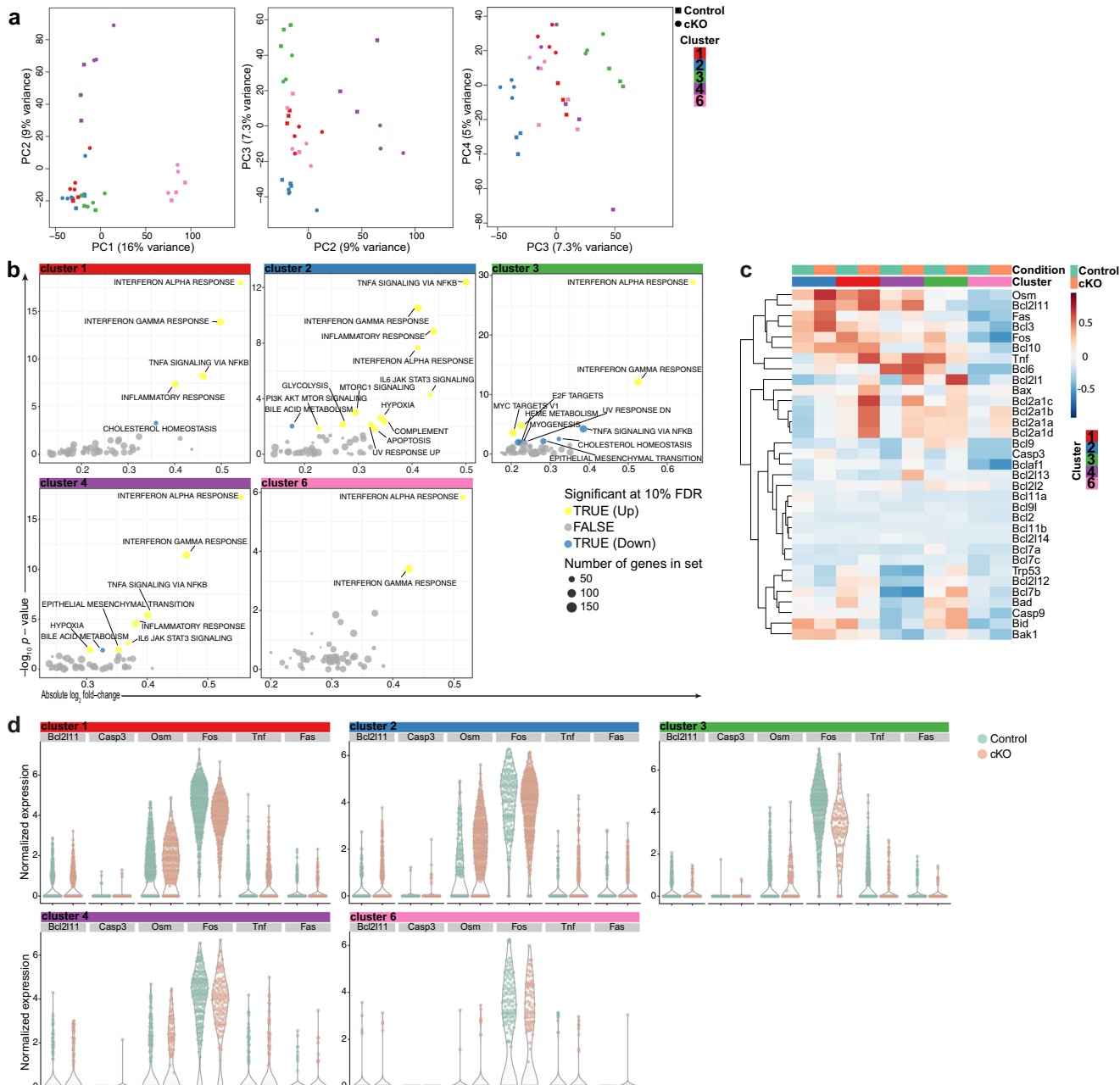

**Fig. 6 Enrichment of apoptosis-related genes in CD98hc cKO cells. a** Gene expression variations between control and cKO cells were retrospectively analyzed and presented in a principal component analysis plots for each cluster. **b** Gene set enrichment analysis indicates enrichment of differential expresses genes in CD98 cKO cells over control cells in indicated signatures per cluster. **c** Heatmap of genes associated with apoptosis were displayed. **d** Expression of *Bcl2l11*, *Casp3*, *Osm*, *Fos*, *Tnf*, and *Fas* by CD98 cKO cells and control cells per individual cluster. The experiment was done once (scRNA-seq) with four biological replicates in each group.

present), and the presence or absence of goblet cell depletion (0: absent; 1: present). Each feature score was summed up to a maximum possible score of 11. Histological scores were assessed in a blinded fashion by two independent investigators, and the mean histological score was determined for each animal.

**Isolation of bone marrow cells**. After the connective tissues and muscles were removed from prepared femurs and tibia, the bones were opened at the epiphysis. A syringe with a 25-gauge needle was placed into the ends of the opened femurs and tibias. Bone marrow cells were flushed out with RPMI 1640 medium (Sigma). The collected cells were passed through a 70-μm cell strainer to remove cell clumps and bone fragments.

**Colonic lamina propria cell isolation**. The isolated colon was opened longitudinally and washed with PBS to remove debris and mucus. The intestinal epithelium was removed by incubation in 5 mM EDTA in $Ca^{2+}/Mg^{2+}$-free PBS at

37 °C under gentle shaking for 10 min for a total of three incubations. After every incubation cycle, the tubes were vortexed for 30 s, and the tissue pieces were transferred into fresh EDTA/PBS. The colon was washed in PBS to remove residual EDTA. The tissue was cut as small as possible and digested with 0.5 mg/ml Collagenase type VIII (Sigma-Aldrich) and 10 U/ml DNase (Roche) in RPMI 1640 for 20–25 min at 37 °C in a water bath with continuous shaking (200 rpm). Every 5 min, the tubes were vortexed manually for 30 s. Supernatants were collected and passed through a 70-mm cell strainer, and cells were pelleted by centrifugation. The cells were counted and processed for flow cytometry analysis.

**Yolk-sac cell isolation**. The yolk sac was harvested from embryos at E8.5. Embryos were exsanguinated through decapitation in PBS containing 3% fetal calf serum (FCS, Gibco). To obtain a single-cell suspension, the yolk sac was incubated in RPMI 1640 medium containing 1 mg/ml collagenase type VIII, 100 U/ml DNase I, and 3% FCS at 37 °C for 30 min. The digested yolk sac was poured through a

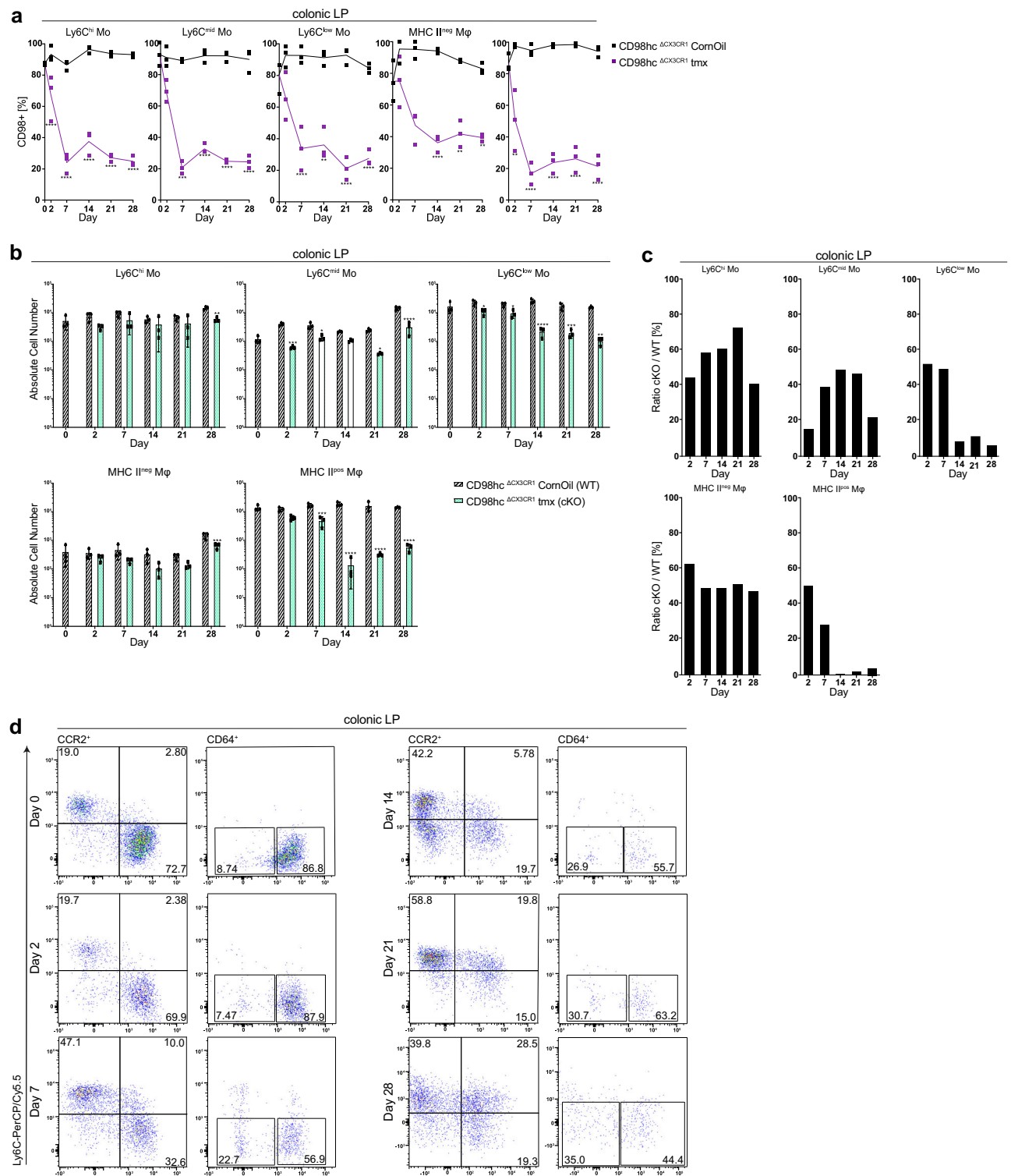

**Fig. 7 Deletion of CD98hc in monocytes and macrophages leads to reduced macrophage numbers in the colonic lamina propria.** Percentage of monocytes and macrophages that express CD98hc and total monocyte and macrophage number in the colonic lamina propria of CD98hc$^{\Delta CX3CR1}$ mice was determined after receiving tamoxifen for 28 days. **a** Percentage of CD98hc$^+$ monocytes and macrophages, and **b** the total monocyte and macrophage numbers in the colonic lamina propria of corn oil- and tamoxifen-treated CD98hc$^{\Delta CX3CR1}$ animals ($n = 3$) are shown as the mean (± SD) and analyzed by two-way ANOVA followed by Sidak's correction; ****$p < 0.0001$. **c** Ratios (cKO/WT cells) of total colonic lamina propria monocytes and macrophages cell numbers between corn oil and tamoxifen-treated CD98hc$^{\Delta CX3CR1}$ at indicated time points. **d** Dot plots were generated by gating on CCR2$^+$CD64$^-$/ CCR2$^+$CD64$^+$ and CCR2$^-$CD64$^+$ colonic lamina propria monocytes and macrophages isolated from tamoxifen-treated CD98hc$^{\Delta CX3CR1}$ animals at indicated time points. Numbers show the percentage of the indicated gates. Experiments performed twice with three biological replicates in each group.

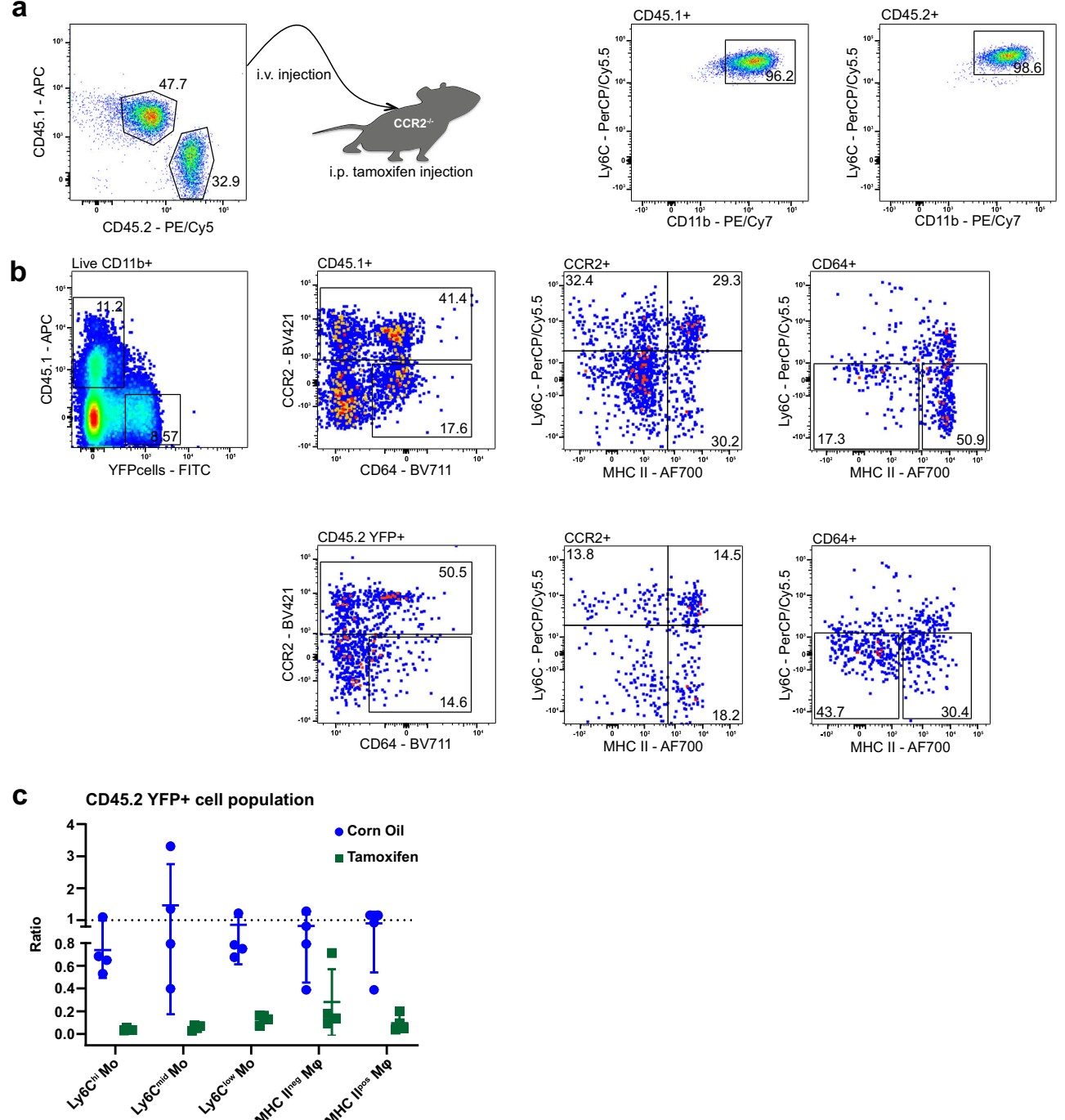

**Fig. 8 Deletion of CD98hc leads to reduced MHCII⁺ macrophages in competitive adoptive BM monocyte transfer experiments.** After isolation of bone marrow monocytes from B6 Ly5.1 (CD45.1⁺) and CD98hc$^{\Delta CX3CR1}$ (CD45.2 YFP⁺), CD11b⁺ monocytes were injected into CCR2$^{-/-}$ recipients. **a** Dot plots of bone marrow CD45.1 and CD45.2 monocytes mixtures before transfer into CCR2$^{-/-}$ animals. CD11b⁺/Ly6C⁺ dot plots were obtained after gating either on CD45.1 or CD45.2 monocytes. **b** Dot plots were generated by gating on live CD11b⁺ colonic lamina propria monocytes and macrophages isolated from tamoxifen-treated CCR2$^{-/-}$ recipients. **c** Ratios between CD45.2 YFP⁺ and CD45.1⁺ WT colonic lamina propria monocytes and macrophages isolated from corn oil- or tamoxifen-treated CCR2$^{-/-}$ hosts 5 days after transfer normalized to input ratios. Each dot indicates one respective host. Experiment was performed once with four biological replicates in each group.

70-μm cell strainer, and erythrocytes were lysed (3–5 min at room temperature with Tris-Lysing buffer (144 mM NH₄Cl, 17 mM Tris)). Cells were counted and processed for flow cytometry analysis.

**Isoflurane anesthesia**. Mice were anesthetized with the inhalation anesthetic isoflurane for the collection of the liver. For the induction of narcosis, mice were placed in a narcosis chamber that was flooded with 2–3% isoflurane in oxygen at

1–2 l/min. Anesthesia was confirmed by assuring a decrease in the respiration rate and by testing for the absence of the pedal withdrawal reflex. For hepatectomy, anesthetized mice were placed under a mask with isoflurane flow, as described above.

**Liver perfusion and liver cell isolation**. The portal vein of an anesthetized (Isoflurane) animal was punctured with a 25-gauge needle. The liver was in situ

perfused with 37 °C prewarmed 10 ml (~7–9 ml/min) liver perfusion medium (Gibco) followed by 37 °C prewarmed 5 ml (~3 ml/min) liver digest medium (Gibco) after cutting the lower vena cava. Following removal of the gallbladder, the liver was placed into a petri dish and cut into small pieces. The tissue pieces were transferred into a 50-ml tube containing 5 ml of liver digest medium and digested for 30 min at 37 °C. Afterward, the digested tissue was poured and mashed through a metal cell strainer to remove connective tissue and centrifuged for 5 min and 500 rpm at room temperature. The supernatant (solution A) and pellet (solution B) were separated into two tubes. The solution A was centrifuged for 5 min and 1400 rpm at room temperature. To solution B, 40 ml of PBS was added and centrifuged for 5 min and 500 rpm at room temperature. The supernatant from solution A was discarded, and the supernatant from solution B was added to the pellet of solution A. After centrifugation for 5 min and 1400 rpm at room temperature, the supernatant was discarded, and the pellet frothed up with 3 ml of PBS/2% FBS supplemented with 0.1% w/v sodium azide and 10 mM EDTA and 3.5 ml 70% Percoll (GE Healthcare) to obtain the "Cell-Percoll-Suspension". A Percoll gradient was prepared and centrifuged for 20 min and 2000 rpm without break. The fat layer on the top was removed, and the interphase, which contains the lymphocytes and erythrocytes as well as the whole upper liquid phase to increase the cell yield, were collected. After the erythrocytes were lysed (3–5 min at room temperature with Tris-Lysing buffer (144 mM $NH_4Cl$, 17 mM Tris)), the cells were counted and processed for flow cytometry analysis.

**Langerhans cell isolation**. After cutting off the mouse ears, the ears are divided into dorsal and the ventral halves, from which the cartilage is removed with forceps. The ears were then placed dermal side down onto PBS containing 2.5 mg/ml dispase II (Sigma) and were incubated for 2 h at 37 °C. The dissociated epidermal sheets are placed in stop medium (2% FCS in PBS) and further transferred into a 50 ml tube with 20 ml RPMI 1640 medium containing 10% FCS and supplemented with 0.05 mM 2-ME, 100 U/ml penicillin, and 100 mg/ml streptomycin. To release the Langerhans cells (LCs), the tube was gently shaking for 30 min at 37 °C in a water bath. The remaining epidermal pieces and cell suspension were filtered through a 70-μm cell strainer and cells pelleted by centrifugation for 5 min and 1400 rpm at 4 °C. The cells were counted and processed for flow cytometry analysis.

**Microglia isolation**. Mouse brain was collected, chopped into small pieces, and subsequently smashed through a 70-μm cell strainer. Afterward, microglia were isolated using the CD11b (Microglia) MicroBeads Kit (Miltenyi Biotec) according to the manufacturer's protocol. The cells were counted and processed for flow cytometry analysis.

**Cardiac macrophage isolation**. Before sacrificing the mice (minimum 5 min in advance), circulating leukocytes were labeled by intravenous injection of 1 μg of anti-mouse CD45-Superbright in a final volume of 200 μl PBS per mouse using an insulin syringe. The chest under the sternum was opened by cutting the ribs, and then the heart removed from the body. The right atrium was cut, and the heart was gently perfused through the apex with 20 ml of cold 1× PBS (~7–9 ml/min). This step was repeated by perfusing the right ventricle. After that, the atria were dissected from the ventricles and discarded. The isolated ventricles were then transferred into 1.5-ml tubes containing 1 mL cold digestion medium (1× PBS with 100 μg/ml collagenase IV (Sigma)), in which the heart was chopped into small pieces. For digestions, the samples were incubated at 37 °C for 45 min with gentle shaking (~30 rpm). Samples were mechanically homogenized by up and down motions through a 1-ml syringe capped with an 18 G needle. Afterward, the samples were transferred into a 50-ml tube through a 70-μm cell strainer to remove the tissue stroma. Cells were pelleted by centrifugation. The supernatant was discarded, and erythrocytes were eliminated by blood cell lysis buffer for ~3 min. Cells were washed with 10 ml FACS buffer. Counted cells were further used for flow cytometry analysis.

**Immunohistochemistry and immunofluorescence**. Cryopreserved biopsies of patients with Crohn's disease or with ulcerative colitis embedded in Tissue-Tek O. C.T. compound (Sakura) were acquired from the Basel inflammatory bowel disease cohort. Immunohistochemistry was performed on 6-μm sections using a polyclonal rabbit anti-human CD98hc (ThermoFisher). Primary antibody (1:50 dilution) binding was detected with an Alexa Flour 647 goat anti-rabbit IgG secondary antibody (1:200) (ThermoFisher). Six-micrometer sections from cryopreserved mouse tissues were fixed in 4% paraformaldehyde for 15 min at room temperature. Afterward, sections were blocked with goat serum in DPBS/0.4% Triton-X-100 (Roth) for 30 min and stained with the primary monoclonal rabbit anti-mouse CD98hc mAb overnight in a humidified container at 4 °C. Primary antibody (1:50) binding was detected with an Alexa Flour 647 goat anti-rabbit IgG secondary antibody (1:200) (ThermoFisher). Sections were counterstained with NucBlue™ Live Cell Stain Ready Probes™ reagent (1 drop of stain in 500 μl 1× PBS) (Molecular Probes, Hoechst 33342 unique formulation) and imaged with a Nikon A1R Nala confocal microscope.

**Bone marrow-derived macrophages**. Murine bone marrow cells were cultured in six-well plates in the RPMI 1640 medium (Sigma), or nonessential amino acid RPMI 1640 medium (BioConcept) without or with supplementation of L-leucine (0.05 g/L) containing 10% FCS and supplemented with 0.05 mM 2-ME, 100 U/ml penicillin, and 100 μg/ml streptomycin. Bone marrow-derived macrophages were generated by adding 20 ng/ml M-CSF (BioLegend) to the cultures. After 7 days, macrophages were either stimulated with 100 ng/ml lipopolysaccharide (LPS) from *Escherichia coli* O111:B4 (Sigma) and 10 ng/ml recombinant mouse IFN-γ (rmIFN-γ; BioLegend) or with 10 ng/ml recombinant mouse IL-4 (rmIL-4; BioLegend) and 10 ng/ml recombinant mouse IL-13 (rmIL-13; BioLegend) for 6 h before cells were analyzed. For conditional deletion of CD98c in vitro, tamoxifen dissolved in DMSO (Roth) was added into the culture during the macrophage generating and during LPS + IFNγ or IL-4 + IL-13 stimulation or D-phenylalanine (Sigma) was added 1 h prior and during LPS + IFNγ or IL-4 + IL-13 stimulation.

**Adoptive monocyte transfer**. After isolation of bone marrow monocytes from B6 Ly5.1 (CD45.1$^+$) and CD98hc$^{ΔCX3CR1}$ (CD45.2 YFP$^+$) animals with the Monocyte Isolation Kit (Miltenyi Biotec) according to the manufacturer's protocol, purified bone marrow CD11b$^+$ Ly6C$^+$ monocytes (1–2 ×10$^6$ cells/mouse) from donors were intravenously transferred into CCR2$^{−/−}$-recipient mice, from which sufficient donor macrophage numbers can be recovered[52]. Bone marrow monocytes from CD45.1$^+$ WT and CD45.2 YFP$^+$ CD98hc$^{ΔCX3CR1}$ animals were mixed in a 1:1 ratio before transfer in the competitive adoptive monocyte transfer experiments. CD45.1$^+$ WT and CD45.2 YFP$^+$ CD98hc$^{ΔCX3CR1}$ cells were intravenously transferred into CD45.2 CCR2$^{−/−}$ corn oil or tamoxifen-treated recipients. Before injection, a small proportion of donor cells were saved for calculating the input ration (IR). These cells were extracellularly stained for CD45.1 (APC-conjugated, 1:100 dilution), CD45.2 (biotin-conjugated, second step reagent PE/Cy5-conjugated streptavidin, 1:100), Ly6C (PerCP/Cy5.5-conjugated, 1:400), and CD11b (PE/Cy7-conjugated, 1:100). Cells were analyzed by flow cytometry, and IR was calculated as IR = number of CD45.2 YFP$^+$/CD45.1$^+$ cells. Donor monocyte-derived cells were recovered from the colonic lamina propria of recipient mice on day 5 after adoptive monocyte transfer and after the first corn oil or tamoxifen treatment. The cells were counted and processed for flow cytometry analysis, and ratios of recovered monocytes and macrophages were normalized to IR.

**Surface staining for flow cytometry**. After isolation of the indicated tissues, cells were washed in PBS, and subsequently stained with fixable viability dye eFluor455UV (1:500 dilution) (eBiosciences) and mAb (Clone 93) (ThermoFisher) directed against the FcγRIII/II CD16/CD32 (0.5 μg mAb/10$^6$ cells) for 20 min at 4 °C. Cells were washed in PBS/2% FBS supplemented with 0.1% w/v sodium azide and 10 mM EDTA, incubated with the relevant mAb for 20 min at 4 °C and washed again twice. When primary antibodies were biotin-coupled antibodies, cells were incubated with Streptavidin-PE/Cy5 (1:800) (BioLegend) for 20 min at 4 °C. Data were acquired with the BD LSRFortessa™ X-20 flow cytometer (BD Biosciences) and analyzed using FlowJo software version 10.5.3 (Tree Star Inc.). Cell sorting was carried out with the BD FACSAria™ III equipment. In all experiments, forward scatter (FSC)-H versus FSC-A was used to gate on singlets, with dead cells excluded using the fluorescence-coupled fixable viability dye. Lineage positive (CD3, CD19, NK1.1, Ly6G, and Ter119) cells were removed from further analysis.

**Intracellular staining for flow cytometry**. After surface staining, cells were resuspended in BD Cytofix/Cytoperm (BD Bioscience) and incubated for 20 min at 4 °C. Cells were washed in Perm/Wash Buffer (BD Bioscience), incubated with the relevant mAb for 20 min at 4 °C, and washed again twice. Data were acquired with a BD FACSAria™ III instrument.

**mAbs for surface and intracellular staining**. For surface staining, we used the following mAbs: biotin-conjugated anti-CD3 145-2C11 (1:400 dilution), biotin-conjugated anti-CD19 6D5 (1:100), biotin-conjugated anti-NK1.1 PK136 (1:400), biotin-conjugated anti-Ly6G 1A8 (1:400), biotin-conjugated anti-Ter119 TER-119 (1:400), biotin-conjugated anti-F4/80 BM8 (1:100), biotin-conjugated anti-CD45.2 104 (1:100), APC-conjugated anti-CD45.1 A20 (1:100), Alexa Fluor 700-conjugated anti–I-A/I-E M5/114.15.2 (1:200), PerCP/Cy5.5-conjugated anti-Ly6C HK1.4 (1:400), PE/Cy7-conjugated anti-CD11b M1/70 (1:100), PE/Cy7-conjugated Annexin V (1:20), PE-conjugated anti-CD98 RL388 (1:800), Brilliant Violet 711-conjugated anti-CD64 X54-5/71 (1:200), APC-conjugated anti-F4/80 BM8 (1:50), PE/Cy5-conjugated anti-F4/80 BM8 (1:100), Pacific Blue-conjugated anti-CD29 HMβ1–1 (1:200), Alexa Fluor 700-conjugated anti-CD3 17A2 (1:100), Brilliant Violet 510-conjugated anti-CD4 GK1.5 (1:100), PerCP or PerCP/Cy5.5-conjugated anti-CD8 53-6.7 (1:100), APC/Fire 750-conjugated anti-CD11c N418 (1:50), APC-conjugated anti-Ly6G 1A8 (1:200), Brilliant Violet 785-conjugated anti-CD19 6D5 (1:50), APC-conjugated anti-CD207 4C7 (1:100), PE/Dazzle-conjugated anti-CD115 AFS98 (1:200), APC-conjugated anti-CD135 A2F10 (1:25), PE/Dazzle-conjugated anti-CD14 Sa14-2 (1:100), PerCP/Cy5.5-conjugated CD81 Eat-2 (1:100), APC/Cy7-conjugated ant-CD45 30-F11 (1:400), SuperBright anti-CD45 30-F11 (1:400) (all BioLegend), APC-conjugated anti-CCR2 475301 (10 μl/Test) (R&D Systems), eVolve 655-conjugated anti-CD45 30-F11 (1:400) (eBioscience), Brilliant Blue 515-conjugated anti-CD117 2B8 (1:200), Brilliant Violet 786-

conjugated anti-CD72 K10.6 (1:100) (all BD Bioscience). For intracellular staining, the following mAbs were used: Alexa Fluor 488-conjugated anti-phospho-S6 ribosomal protein (Ser235/236) D57.2.2E (1:50), unconjugated anti-phospho-p70S6 Kinase (Thr389) 108D2 (1:50), PE-conjugated anti-rabbit IgG (all Cell Signaling) (1:300), unconjugated anti-Zeb2 CUK2 (50-100 µl/Test), DyLight 405-conjugated anti-rabbit IgG (1:100) (Jackson ImmunoResearch). For detecting mouse/human ZEB2 protein (also known as SIP1), we used a homemade antibody that has been validated for specificity using siRNA and western blotting, immunohistochemistry, immune-precipitation, and immunofluorescence[53,54].

**Quantitative real-time PCR**. RNA was extracted from colonic tissue using the Direct-zol MiniPrep Kit (Zymo Research) according to the manufacturer's protocol. In column, DNase I treatment removed genomic DNA and reversed transcribed into cDNA using Superscript RT III (Invitrogen), according to the manufacturer's instructions. Quantitative real-time PCR (qRT-PCR) was carried out in 384-well plates using gene-specific primers and either SsoFast EvaGreen Supermix (Bio-Rad) and run on a Bio-Rad CFX384 cycler or with a QuantiNova SYBR Green PCR kit (QIAGEN) and run on an ABI ViiA 7 cycler. All reactions were run in triplicates. The results were analyzed by the QuantStudio™ Real-Time PCR System Version 1.3 (Applied Biosystems by ThermoFisher). Samples were normalized to the expression of glyceraldehyde 3-phosphate dehydrogenase (*Gapdh*) or Actin-β (*Actb*) by calculating $2^{(-deltaCt)}$. Supplementary Table 5 shows the sequences of primers used for amplification.

**Single-cell RNA sequencing**. For scRNA-seq, lamina propria cells positive for Ccr2 and/or Cd64 were sorted from four corn oil-treated (control) and four tamoxifen-treated (cKO) CD98hc$^{\Delta CX3CR1}$ mice. Cell suspensions volumes aiming at a targeted recovery of 3000 cells were loaded on the wells of a 10× Genomics Chromium Single Cell Controller (one well per mouse replicate). The Single Cell 3' v2 Reagent Kit (10× Genomics) was used according to the manufacturer's instructions for single-cell capture, cDNA and library preparation. Single-cell libraries were then sequenced on one flow-cell of an Illumina NexSeq 500 machine at the Genomics Facility Basel of the ETH Zurich (with 58 nt-long R2 reads). Read quality was determined with the FastQC tool (version 0.11.5). For sample and cell demultiplexing, read alignment to the mouse mm10 genome assembly with STAR, and generation of a read count table of the sequencing files we used the Cell Ranger software (version 2.1.0). Default settings and parameters were used, except for the version of STAR updated to 2.5.3a, and the STAR parameters *outSAMmultNmax* set to 1 and *alignIntronMax* set to 10000. The reference transcriptome "refdata-cellranger-mm10-1.2.0", provided by 10× Genomics and based on Ensembl release 84[55], was used (available at http://cf.10xgenomics.com/supp/cell-exp/refdata-cellranger-mm10-1.2.0.tar.gz). Because the mouse strain includes a fluorescent reporter gene, a generic EYFP sequence obtained from https://www.addgene.org/browse/sequence_vdb/6394/ was added to the reference transcriptome before mapping. Samples were merged with the "cellranger aggregate" procedure without down-sampling.

Further analysis was performed starting from the UMI counts matrix using the scran (version 1.8.4) and scater (version 1.8.4)[56] Bioconductor packages, following mostly the steps recommended in the simpleSingleCell Bioconductor workflow (version 1.2.1)[57].

Based on the bimodal distributions observed across cells, cells with $\log_{10}$ library sizes less than 2.8 (i.e., a minimum of 630 reads), with $\log_{10}$ total number of features, detected <2.6 (i.e., a minimum of 399 genes detected), with >5% of UMI counts attributed to the mitochondrial genes[58], or with any read attributed to the Hemoglobin genes were excluded. Low-abundance genes with average $\log_2$ CPM (counts per million reads) values <0.005 were not considered.

The raw UMI counts were normalized with the size factors that have been estimated from pools of cells to avoid the dominance of zeros in the matrix[57,59]. To distinguish between genuine biological variability and technical noise, a mean-dependent trend was fitted across expression variances values of endogenous genes[60] (*trendVar* function of the scran package with loess trend and span of 0.05 to better fit the sparse data). The fitted technical noise was subtracted, and the residual component of the gene variance was used to denoise the principal component analysis with the *denoisePCA* function of the scran package. A t-stochastic neighbor embedding (t-SNE) was built with a perplexity of 30 using the top 500 most variable genes and the denoised expression matrix as input.

Clustering of cells into putative subpopulations was performed on normalized log-counts values using hierarchical clustering on the Euclidean distances between cells (with Ward's criterion to minimize the total variance within each cluster; package cluster version 2.0.7-1). Cell clusters were identified by applying a dynamic tree cut (package dynamicTreeCut, version 1.63-1), which resulted in nine clusters. The *findMarkers* function of the scran package, which fits a linear model to the expression values for each gene using the limma framework, was used for the identification of marker genes specific for each cluster.

The R package SingleR (version 1.0.0) was used for reference-based annotation of the cell type of cells and identification of likely contaminants in our data set[31]. We used the Immunological Genome Project (ImmGen) mouse microarray data set[61] as a reference, and retained the main reference cell types for plotting. An enrichment approach was also performed to identify the cell type of each of the clusters, based on the same reference, and gave similar results. Briefly, read counts

of control cells of each cluster were summed to create nine pseudo-bulk samples[32]. Based on the clearly bimodal distribution of expression levels in these samples, and the reference samples, genes were classified as expressed or not-expressed in each sample. The significance of the overlap of expressed genes in each pseudo-bulk sample vs. all reference samples was assessed using a Fisher's exact test. Up to ten top significant reference samples, sorted by decreasing odds ratio, are listed in Supplementary Data 4.

Differential expression between cKO and control cells stratified by differentiation stage was performed by summing the UMI counts of cells from each sample in each cluster when at least 20 cells could be aggregated. This resulted in a total of 32 aggregated samples, and at least 3 replicates per condition for each cluster. The genes were filtered to keep those with CPM (counts per million reads sequenced) values higher than 1 in at least three samples, and detected in at least 20 individual cells. The aggregated samples were then treated as bulk RNA-seq samples[32]: the package edgeR (version 3.24.2)[62] was used to perform TMM normalization[63], and to test for differential expression with the generalized linear model (GLM) framework. Genes with a false discovery rate lower than 5% were assumed as differentially expressed. Gene set enrichment analysis was done with the function camera[64] using the default parameter value of 0.01 for the correlations of genes within gene sets, on gene sets from the Hallmark collection of the Molecular Signature Database (MSigDB, version 6.0)[65,66]. We considered only sets containing more than ten genes, and gene sets with a false discovery rate lower than 5% as significant.

Following, the building of a self-organizing map (SOM) and its minimal spanning tree was performed using the FlowSOM Bioconductor package (version 1.14.0)[67,68], using the 30 first principal components of the denoised principal component analysis as input, and a 9 × 9 grid.

Two-dimensional cell densities were calculated with the kde2d function of the MASS package (version 7.3-50). Differential cell densities on pairs of principal components were calculated as the $\log_2$ of the ratio of the density of cKO cells over the density of control cells (after a prior count of 1e-03 was added to the density estimates).

Remaining statistical analysis on the expression data set analysis and plotting were performed using the R software (version 3.5.1).

**Scanning electron microscopy**. After sorting of Ly6C$^{high}$, Ly6C$^{mid}$, and Ly6C$^{low}$ monocytes and MHCII$^{-}$ and MHCII$^{+}$ macrophages, the respective cell suspension was dropwise added to polylysine-coated round 12-mm glass coverslips and incubated for 15 min. The sample-loaded glass coverslips were once submerged in A. dest. and subsequently dehydrated by an ascending alcohol series of 30%, 50%, 70%, 90%, and two times 100% EtOH each for 10 min. Dehydrated samples were transferred in liquid $CO_2$ into the Autosamdri®-815 (Tousimis) and critical point dried. Dried samples were glued on an aluminum holder and coated with 20-nm gold in the sputter vacuum coater (Leica). Samples were imaged by the FEI Nova Nano SEM 230.

**Statistics and reproducibility**. The data were analyzed with GraphPad Prism software (version 7.03), and are presented as dot plots in which the median of each experimental group is presented in addition to the individual samples. Statistical significance was calculated using the Mann–Whitney *U* test for two groups or using the Kruskal–Wallis test followed by the Dunn's correction test for multiple comparisons. When the data are presented as a time course, the arithmetic mean ± SD is shown. Statistical significance was calculated using two-way ANOVA with the Sidak's correction. Outliers were identified with the Grubb test during the analysis of data acquired from samples from the Swiss IBD Cohort only. The p-values are indicated as follows: $*p \leq 0.05$, $**p \leq 0.01$, $***p \leq 0.001$, and $****p \leq 0.0001$.

**Reporting summary**. Further information on research design is available in the Nature Research Reporting Summary linked to this article.

## Data availability

The scRNA-seq data set is available in the GEO database under accession GSE126574. The Source data underlying the graphs in figures are provided in Supplementary Data 5. Other relevant data are available from the authors upon request.

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

## Acknowledgements
We acknowledge Pawel Pelczar (Center for Transgenic Models, University of Basel) for help with embryo transfers, and Markus Dürrenberger and Susanne Erpel (Nano Imaging Lab, University of Basel), for support with scanning electron microscopy. Claudia Cavelti (Department of Biomedicine, University of Basel) kindly provided the CCR2$^{-/-}$ mice. Gabriela Kuster Pfister (Department of Biomedicine, University of Basel) and Gregor Hutter (Department of Biomedicine, University of Basel) gave advise in cardiac macrophage isolation and microglia isolation. Gennaro de Libero (Department of Biomedicine, University of Basel) and Udo Markert (Placenta-Lab, Friedrich-Schiller-University Jena) for discussions. Christian Beisel (Genomics facility Basel, ETH Zürich) helped with single-cell RNA sequencing. Calculations were performed at sciCORE (http://scicore.unibas.ch/) scientific computing center at the University of Basel. The SNSF grant 310030_175548 supported J.H.N., the SNSF grant 31AC-0_194654 to J.H.N. supported open access publication, and the SNSF grant 33CS30-148422 supported the Swiss IBD Cohort Investigators.

## Author contributions
Ph.W., B.K., and H.M. performed experiments and J.R. bioinformatic analysis. A.E.S. provided the unconjugated anti-Zeb2 (CUK2) antibody, P.H. and the Swiss IBD Cohort Investigators provided patient samples, and H.T. and M.I. the cryopreserved CD98hc$^{flox/flox}$ embryos. J.H.N. analyzed and interpreted the data. Ph.W., B.K., H.M., C.K.A., A.E.S., J.R., H.T., M.I., and J.H.N. discussed the data and prepared the paper.

## Competing interests
The authors declare no competing interests.

## Additional information

## Swiss IBD Cohort Investigators

Karim Abdelrahman[7], Gentiana Ademi[8], Patrick Aepli[9], Claudia Anderegg[10], Anca-Teodora Antonino[11], Eva Archanioti[12], Eviano Arrigoni[13], Diana Bakker de Jong[2], Bruno Balsiger[14], Polat Bastürk[2], Peter Bauerfeind[15], Andrea Becocci[16], Dominique Belli[16], José M. Bengoa[13], Luc Biedermann[17], Janek Binek[18], Mirjam Blattmann[17], Stephan Boehm[19], Tujana Boldanova[2], Jan Borovicka[8], Christian P. Braegger[20], Stephan Brand[8], Lukas Brügger[21], Simon Brunner[2], Patrick Bühr[20], Sabine Burk[17], Bernard Burnand[22], Emanuel Burri[23], Sophie Buyse[24], Dahlia-Thao Cao[25], Ove Carstens[21], Dominique H. Criblez[9], Sophie Cunningham[13], Fabrizia D'Angelo[26], Philippe de Saussure[13], Lukas Degen[2], Joakim Delarive[27], Christopher Doerig[28], Barbara Dora[17], Susan Drerup[29], Mara Egger[22], Ali El-Wafa[30], Matthias Engelmann[31], Jessica Ezri[12], Christian Felley[32], Markus Fliegner[33], Nicolas Fournier[22], Montserrat Fraga[12], Yannick Franc[22], Remus Frei[8], Pascal Frei[34], Michael Fried[17], Florian Froehlich[35], Raoul Ivano Furlano[36], Luca Garzoni[37], Martin Geyer[38], Laurent Girard[13], Marc Girardin[39], Delphine Golay[22], Ignaz Good[40], Ulrike Graf Bigler[21], Beat Gysi[41], Johannes Haarer[8], Marcel Halama[42], Janine Haldemann[14], Pius Heer[43], Benjamin Heimgartner[21], Beat Helbling[34], Peter Hengstler[18], Denise Herzog[44], Cyrill Hess[9], Roxane Hessler[28], Klaas Heyland[45], Thomas Hinterleitner[46], Claudia Hirschi[31], Petr Hruz[2], Pascal Juillerat[21], Stephan Kayser[47], Céline Keller[48], Carolina Khalid-de Bakker[2], Christina Knellwolf(-Grieger)[8], Christoph Knoblauch[49], Henrik Köhler[10], Rebekka Koller[20], Claudia Krieger(-Grübel)[8], Patrizia Künzler[8], Rachel Kusche[10], Frank Serge Lehmann[43], Andrew J. Macpherson[21], Michel H. Maillard[12,48], Michael Manz[2], Astrid Marot[22], Rémy Meier[50], Christa Meyenberger[8], Pamela Meyer[8], Pierre Michetti[12,48], Benjamin Misselwitz[17], Patrick Mosler[51], Christian Mottet[52], Christoph Müller[53], Beat Müllhaupt[17],

Leilla Musso[22], Michaela Neagu[54], Cristina Nichita[55], Jan H. Niess[2], Andreas Nydegger[12,48], Nicole Obialo[17], Diana Ollo[32], Cassandra Oropesa[32], Ulrich Peter[45], Daniel Peternac[56], Laetitia Marie Petit[32], Valérie Pittet[22], Daniel Pohl[17], Marc Porzner[57], Claudia Preissler[58], Nadia Raschle[17], Ronald Rentsch[59], Sophie Restellini[32], Alexandre Restellini[39], Jean-Pierre Richterich[9], Frederic Ris[32], Branislav Risti[60], Marc Alain Ritz[61], Gerhard Rogler[17], Nina Röhrich[8], Jean-Benoît Rossel[22], Vanessa Rueger[20], Monica Rusticeanu[21], Markus Sagmeister[62], Gaby Saner[14], Bernhard Sauter[63], Mikael Sawatzki[8], Michael Scharl[17], Martin Schelling[8], Susanne Schibli[64], Hugo Schlauri[65], Dominique Schluckebier[32], Sybille Schmid(-Uebelhart)[21], Daniela Schmid[49], Jean-François Schnegg[66], Alain Schoepfer[12,48], Vivianne Seematter[22], Frank Seibold[14], Mariam Seirafi[67], Gian-Marco Semadeni[8], Arne Senning[20], Christiane Sokollik[64], Joachim Sommer[22], Johannes Spalinger[9,64], Holger Spangenberger[68], Philippe Stadler[69], Peter Staub[70], Dominic Staudenmann[9], Volker Stenz[71], Michael Steuerwald[61], Alex Straumann[43], Bruno Strebel[21], Andreas Stulz[9], Michael Sulz[8], Aurora Tatu[21], Michela Tempia-Caliera[72], Amman Thomas[73], Joël Thorens[74], Kaspar Truninger[75], Radu Tutuian[21], Patrick Urfer[76], Stephan Vavricka[17], Francesco Viani[77], Jürg Vögtlin[61], Roland Von Känel[17], Dominique Vouillamoz[78], Rachel Vulliamy[22], Paul Wiesel[55], Reiner Wiest[21], Stefanie Wöhrle[9], Tina Wylie[79], Samuel Zamora[39], Silvan Zander[80], Jonas Zeitz[17] & Dorothee Zimmermann[8]

[7]Clinique de Montchoisi, Lausanne, Switzerland. [8]Kantonsspital St. Gallen, St. Gallen, Switzerland. [9]Kantonsspital Luzern, Luzern, Switzerland. [10]Kantonspital Aarau, Klinik für Kinder und Jugendliche, Aarau, Switzerland. [11]Hôpital Riviera–Site du Samaritain, Vevey, Vaud, Switzerland. [12]Service of Gastroenterology and Hepatology, Department of Medicine, Centre Hospitalier Universitaire Vaudois and University of Lausanne, Lausanne, Switzerland. [13]GI private practice, Geneva, Switzerland. [14]Gastroenterologische Praxis, Bern, Switzerland. [15]Department Gastroenterology and Hepatology, Stadtspital Triemli, Zurich, Switzerland. [16]Department of Pediatric, Geneva University Hospital, Geneva, Switzerland. [17]Department of Gastroenterology and Hepatology, University Hospital Zurich, University of Zurich, Zurich, Switzerland. [18]Gastroenterologie am Rosenberg, St. Gallen, Switzerland. [19]Spital Bülach, Bülach, Zurich, Switzerland. [20]University Children's Hospital, Zurich, Switzerland. [21]Department of Visceral Surgery and Medicine, Bern University Hospital, University of Bern, Bern, Switzerland. [22]Institute of Social and Preventive Medicine (IUMSP), Lausanne University Hospital, Lausanne, Switzerland. [23]Department Gastroenterology, Kantonsspital Liestal, Liestal, Switzerland. [24]GI private practice, Yverdon-les-Bains, Switzerland. [25]Hôpital Neuchâtelois, La Chaux-de-fonds, Neuchâtel, Switzerland. [26]Department Gastroenterology and Hepatology, Geneva University Hospital, Geneva, Switzerland. [27]GI private practice, Lausanne, Switzerland. [28]Clinique Cecil, Lausanne, Switzerland. [29]Schulthess Clinic, Zurich, Switzerland. [30]GI private practice, La Chaux-de-Fonds, Switzerland. [31]Gastropraxis Luzern, Luzern, Switzerland. [32]Centre de Gastroentérologie Beaulieu SA, Geneva, Switzerland. [33]Medical Center Sihlcity, Zurich, Switzerland. [34]Gastroenterologie Bethanien, Zurich, Switzerland. [35]Hospital of the Canton of Jura, Porrentruy And Delémont, Jura, Switzerland. [36]Universitäts-Kinderspital beider Basel (UKBB), Basel, Switzerland. [37]Clinique des Grangettes, Geneva University Hospital, Genève, Switzerland. [38]GI private practice, Wettingen, Aargau, Switzerland. [39]Groupe Médical d'Onex, Onex, Switzerland. [40]Spital Walenstadt, Walenstadt, St. Gallen, Switzerland. [41]GI private practice, Reinach, Switzerland. [42]Aerztehaus Fluntern, Zurich, Switzerland. [43]GI private practice, Olten, Switzerland. [44]HFR Hôpital fribourgeois–Pédiatrie, Fribourg, Switzerland. [45]KSW Kantonsspital Winterthur Kinderklinik, Winterthur, Switzerland. [46]GI private practice, Zurich, Switzerland. [47]GI private practice, Luzern, Switzerland. [48]Gastroenterology La Source-Beaulieu, Lausanne, Switzerland. [49]Kantonsspital Nidwalden, Stans, Nidwalden, Switzerland. [50]AMB – Arztpraxis MagenDarm Basel, Basel, Switzerland. [51]Kantonsspital Graubünden, Chur, Switzerland. [52]GI private practice, Sion, Switzerland. [53]Division of Experimental Pathology, Institute of Pathology, University of Bern, Bern, Switzerland. [54]Spital Tiefenau, Bern, Switzerland. [55]Centre médical d'Epalinges, Epalinges, Switzerland. [56]Spital Waid, Zurich, Switzerland. [57]Spital Lachen, Lachen, Switzerland. [58]Kantonsspital Olten, Olten, Switzerland. [59]GI private practice, St. Gallen, Switzerland. [60]GI practice, Dietikon, Switzerland. [61]GI practice, Liestal, Switzerland. [62]GI private practice, Heerbrugg, Switzerland. [63]Klinik Hirslanden Zürich, Zurich, Switzerland. [64]Kinderklinik Bern, Bern University Hospital, Bern, Switzerland. [65]Derby Center, Wil, Switzerland. [66]GI private practice, Montreux, Switzerland. [67]Clinique La Colline, Geneva, Switzerland. [68]Kantonsspital Wolhusen, Wolhusen, Switzerland. [69]GI private practice, Payerne, Switzerland. [70]Spital Heiden Appenzell Ausserrhoden, Heiden, Switzerland. [71]Kantonsspital Münsterlingen, Münsterlingen, Switzerland. [72]Clinique des Grangettes, Chêne-Bougeries, Switzerland. [73]GI private practice, Waldkirch, St. Gallen, Switzerland. [74]GI private practice, Yverdon, Switzerland. [75]GI private practice, Langenthal, Switzerland. [76]Hirslanden Klinik Aarau, Gastro Zentrum, Aarau, Switzerland. [77]Private practice, Vevey, Switzerland. [78]Private practice, Pully, Switzerland. [79]Infirmière de Recherche chez CHUV Lausanne University Hospital, Lausanne, Switzerland. [80]Spital Limmattal, Schlieren, Switzerland.

