## [Peer Review File · Communications Biology]

Reviewers' comments:

Reviewer #1 (Remarks to the Author):

The data presented by Wuggenig and co-workers aims to assess the role of CD98 in the maturation of monocytes and macrophages in the gut mucosa. The authors use the well-described Cx3cr1-Cre-ERT-YFP mouse to specifically delete CD98 in this lineage. Collectively, the results show that loss of CD98 alters monocyte to macrophage differentiation and this results in diminished severity of chemically-induced colitis. These results will be of interest to the mucosal immunology field. However, there are major concerns that the authors need to address.

- In Figure 1, the gate used to select CD64+ cells is very broad. The authors should show a staining control for the use of such a wide gating strategy.
 - Unbiased analysis using tSNE plots is a nice way to display these data. However, in order for the reader to properly interpret these, data showing the expression of each marker used to generate the tSNE analysis should be shown either in the main figure or as supplementary data. This is needed to understand which markers are generating the particular clusters shown in the BM and colon. The use of 'heatmap' tSNE are good for this purpose.
 - The overlay tSNE is difficult to visualise due to the rarity of e.g. P1. Also, given that so-called 'P5' cells dominate the macrophage compartment, why are grey cells not the dominant population on the tSNE plot? It would be more informative to have individual tSNE plots for each population. Having CD98 expression on the tSNE is rather meaningless given that most populations are uniformly positive.
 - The identification of Kupffer cells (KC) is substandard. KC are the most abundant myeloid cell in the liver but in this manuscript they form only a small population. Although KC are not the focus on the manuscript, conclusions are based on these data and it is not clear if these are true KC, especially because CD11c+ and CD11c- subsets are identified which is not a conventional way to define these cells. In particular, the CD11c+ 'KC' do not fit the characterisation as KC given they lack high expression of F4/80 and low expression of CD11b. Another major concern is that the authors report an effect of tamoxifen administration on CD98 expression amongst these populations in liver, which is unexpected given that bona fide KC (like Langerhans cells) do not express CX3CR1 in the adult (Yona et al. 2013, Immunity, Chorro et al. 2009, JEM). The authors should either repeat these experiments and use a refined isolation protocol or remove reference to KC and instead state that these data analyse liver myeloid cells, which will likely include hepatic macrophages. As the data stand, reference to KC is not justified.
 - The authors should comment on the fact that MHCII- macrophages fail to delete CD98 during colitis. The authors also need to discuss what MHCII- CD64+ cells are, how they relate to MHCII+ macrophages and the potential reasons why they might behave differently.
- In Figure 4 the order of the results does not follow the order of elements e.g. Fig. 4d is mentioned first.
- The data in Figure 4 needs to be presented in a more logical manner.
 - o The exact gating strategy for the sorting for scRNAseq should be shown. My concern about the broad CD64 gate is highlighted here, where many 'contaminating' clusters have been identified.
 - o Confirmatory analysis must be done to show the transcriptional output matches the input (i.e. there is not selective sequencing of a particular subset or from a particular source). Expression of known markers e.g. CCR2, CD64, H2.Aa, CD74, Csf1r, CD68, Lys2 (Macrophages); Cd209a, Napsa (DC), S100A9 (neutrophils) etc would validate that all cells belong to the macrophage lineage. This would be best shown on tSNE plots for each marker. This is important given the authors identify 4 subsets that

seem unrelated. The nature of these data allows identification of positive markers of these clusters (5, 7, 8 and 9). This data should be presented to provide justification for the exclusion of these subsets. scRNAseq provide unparalleled ability to identify novel subsets, therefore full justification for exclusion of subsets is essential.

o The authors state in their introduction that Tim4 and CD4 expression can be used to identify long(er)-lived macrophages in the gut mucosa. To align these data with existing data in the field, the authors should show where CD4 and Tim4 align on their cluster analysis.

o As well as the merged tSNE in Fig. 4b, the same plots of just cells from tamoxifen treated versus corn oil treated mice need to be shown. This will allow the reader to assess if cells from each condition contribute to all of the clusters. A graph showing the contribution of WT vs CD98 KO cells to each cluster (e.g. SFig. 2f) should be shown in the main figure.

o Can the authors confirm that all mice used for the scRNAseq analysis were of the same sex? This is important given that Xist is highlighted as a defining gene in Fig. 4a.

- The data in Supp. Figure 3c should not be shown as a line graph as these are not repeated measured from the same individuals. Also, given the variability in cell numbers from colonic digests, the data should always be compared with a control mouse colon digested on the same day. These data need to be added.
- FACS plots should accompany the data presented in Fig. 6B. The visual of disruption of the monocyte differentiation process will aid the reader.
- It is unclear what is being shown in Figure 6C. If it is a ratio (of KO/WT cells) of the proportion of each subset, then the exact details used to calculate this is need in the figure legend. Also, these data reveal that there is a 3-7 fold increase in the proportion of MHCII⁻ cells in the absence of CD98, which parallels the loss of MHCII⁺ macrophages. This could suggest that loss of CD98 changes the fate of monocytes to a MHCII⁻ macrophage rather than MHCII⁺ cell. Indeed, the absolute number of this subset is increased with tamoxifen treatment (i.e. CD98 deletion).
- The data in Figure 7 would seem more appropriate in Figure 3.
- That deletion of CD98 leads to attenuated colitis seems counterintuitive given that CCR2-dependent monocytes appear to drive chemical colitis and that these cells accumulate in CD98CX3CR1 mice. This needs to be discussed. Indeed, the discussion as a whole is too brief and fails to discuss much of the results.

The authors should state in each figure legend how many times a particular experiment has been performed to allow reproducibility to be assessed.

Reviewer #2 (Remarks to the Author):

This is a compelling report that addresses macrophage CD98 in intestine using Cx3cr1-CreER for inducible deletion of the heavy chain of CD98 in mice. This Cre proved to be usefully specific, because, as opposed to colonic lamina propria monocytes and macrophages, at least in the colon, non-myeloid lineages that express CD98 did not have decreased expression of CD98 with tamoxifen. Furthermore, the mice were protected from DSS-induced colitis with CD98 deletion in Cx3cr1-expressing cells. The expression of apoptosis pathway genes in KO cells found by single cell transcriptomics aligns well with the finding of fewer macrophages. Thus, the report advances knowledge of the role of CD98 in macrophage ontogeny in colitis, and the clinical data are of interest. Thus, the findings represent a noteworthy contribution. The major shortcoming is that the link between CD98 and protection from apoptosis is not made.

The authors mention data not shown that are quite intriguing: "Furthermore, feeding animals with

10% higher amino acid chow (5% more leucine and 5% more isoleucine) to induce an increased inflammation on a CD98 dependent manner, indicated a tendency to escalate DSS-induced inflammation (data not shown) compared to conventional chow. These results indicate that the loss of CD98 in CX3CR1+ macrophages attenuated DSS-induced colitis in mice." It would be worthwhile to present these data and include them as a supplement, even if the significance is at a trend level. Were there more macrophages with extra amino acids? In this connection, mTOR mediates inflammatory activation of macrophages (PMID:24280772, Byles et al. Nat Comm), in a manner that is mediated by transporter Slc7a5 (PMID 29422900, Yoon et al Frontiers in Immunology). In vitro or in vivo, could colonic macrophages with and without CD98 be treated with leucine with measurement of M1 markers and mTOR pathway activation? In this manner, could a link between CD98, protection against apoptosis, and mTOR be made? In sum, some mechanistic insight into the anti-apoptotic pathways regulated by CD98 should be feasible within a short time frame and would be a valuable addition to this story.

The single cell data are intriguing but cellular annotation is marker-based. Existing free, publicly available annotation tools (for example, among several options: PMID: 30643263, Aran et al. Nat Imm) that use reference-based annotation for unbiased identification of cell types will confirm contaminating cell types and tighten the analysis of putative cell types of interest (monocytes and macrophages).

Reviewer #3 (Remarks to the Author):

The Branched-Chain Amino Acid Transporter CD98 Heavy Chain Facilitates the Development of Colonic Macrophages Associated with Decreased Apoptosis in Macrophage Progenitors
Philipp Wuggenig, Berna Kaya, Hassan Melhem, Korcan Ayata, Swiss IBD Cohort Investigators, Petr Hruz, Hideki Tsumura, Morihito Ito, Julien Roux, Jan Hendrik Niess

Summary

The study is aimed at investigating the importance of CD98 on monocyte differentiation into gut macrophages and implications on the pathogenesis of IBD. They show that mononuclear phagocytes and their progenitors express CD98hc in steady state as well as in inflamed conditions. They successfully established a mouse model, CD98hcΔCX3CR1, that silences the expression of CD98 specifically in cLP monocytes and macrophages upon tamoxifen injection and showed that loss of CD98 was associated with attenuated colitis in this model. The authors indicate that CD98 is required for the differentiation of Ly6Chigh monocytes into gut macrophages.

Overall, the manuscript is original and relevant and addresses an important question of the immunological basis of IBD. The introduction however, should be reviewed to provide a clearer description of the model/ rationale. The discussion section should also be reviewed and expanded to further discuss the findings presented and their significance; for instance, in the second paragraph of the discussion (lines 306-313) when the authors describe the impact of silencing CD98, their importance should be better contextualized. The discussion should also include a deeper discussion of the scRNAseq/developmental trajectory findings, including the flow cytometric assessments designed to verify the patterns identified in the scRNAseq analysis.

Major Suggestions

1. The authors should further detail the "monocyte waterfall" in the introduction, including the importance of CX3CR1 as a marker. They should also refer to supplementary fig. 5 in the text and add the marker CCR2 in this schematic representation of the "monocyte waterfall" development (Supplementary Fig. 5);
2. The selection of the CX3CR1CreER mice to generate the CD98hcKO mice should also be expanded

- in the introduction; also consider including a schematic figure to summarize the model as part of the supplementary material;
3. The authors mention in the introduction that long-lived lamina propria macrophages express Tim-4 and surface CD4, therefore they should include these markers in their analysis/discussion;
 4. Supplementary Fig. 2a: the frequency of CD98 was higher in B cells and neutrophils of CD98^{hcflox/flox} and CD98^{hcΔCX3CR1} mice, respectively, than in their controls (corn oil). What are the authors thoughts on these observations?
 5. Supplementary Figure 2a: 10% of macrophages that remain after Tamoxifen treatment. Is this an indication that recombination was incomplete? Please clarify and include the gating strategy for each panel of this Figure (ie. Are the macrophages pre-gated on CX3CR1?).
 6. The authors also state that feeding animals with 10% higher amino acid chow (5% more leucine and 5% more isoleucine) indicated a tendency to escalate DSS-induced inflammation (data not shown) compared to conventional chow. The results should be included as part of the supplementary material. What is the impact of this diet in the frequency/density of CX3CR1^{high} gut resident macrophages and the CXCR1^{int} mononuclear phagocytes?
 7. Methods for liver perfusion: Please expand and add detail to the the methods to include the dose of isoflurane, the rate of perfusion (ml/min), whether there was an in situ perfusion with pre-warmed media as the first step of the two step collagenase perfusion. Please detail the perfusion rate of step 2 (with the liver digest medium). Please fully describe the duration rate and temperature of the 5ml perfusion step.
 8. Figure 4: To assess the quality of single-cell libraries, more sequencing details should be provided. Including the number of UMI detected per cell, the percentage of reads aligned to the transcriptomes, introns, ribosomes, and mitochondria, and the sequencing saturation. Please include 10x Genomics web summaries for each sample in the supplementary data. Please also include the ranked list showing the DE genes for each of the 9 clusters as supplementary data.
 9. Figure 4: For clarity, please show the sort strategy utilized prior to single cell RNA sequencing in the supplementary material. The authors state CCR2⁺/CD64⁺, does this mean either/or, or both.
 10. Figure 4B: It is difficult to differentiate between control and Tamoxifen treated for the map of all 9 clusters. Please show a cluster map in which all control cells are one color and all Tamoxifen treated cells are a different color. It appears that there might be a higher frequency of cluster 2 cells in the KO vs the control mice. Please comment on any differences in the frequency of cells in each cluster for control vs KO.
 11. Figure 4B: Please annotate Clusters 5, 7, 8, and 9 did not seem to be logically related to other clusters using CIBERSORT or GSEA (Evaluation of methods to assign cell type labels to cell clusters from single-cell RNA-sequencing data JJ Diaz-Mejia et al., 1000Research, 2019).
 12. Figure 6A: Why do the author observe and increase in CD98⁺ MHC II^{neg} macrophages between days 14 and 28 of daily tamoxifen treatment?
 13. Figure 6C: It is unclear what is being shown in the figure itself or the legend. Please clarify and show the intra group vs inter group variation, with all data points and statistics (stated as a mean).
 14. Figure 7: Please also evaluate the CD98 expression in healthy individuals as a baseline comparator.

Minor Issues

1. Figure 7D is described in the figure legend but not displayed in Figure 7 as it has been moved to Supplementary Figure 5.
2. Figure 1F: state the difference between P4 and P5 in the figure (MHCII^{neg} and MHCII^{pos});
3. Line 229: refer to Supplementary Fig. 2d and e in the text;
4. Line 273: refer to Fig. 6b in the text;
5. Line 44: followed "by";
6. Line 86: remove "in";

7. Line 346: Figure 7D is not shown and should be removed from the text (should it read Supplementary Fig. 5?).
8. Figures should be formatted to reduce white space (particularly Figure 1).

Point-to-Point response

(manuscript COMMSBIO-19-0371)

Reviewer #1 (Remarks to the Author):

The data presented by Wuggenig and co-workers aims to assess the role of CD98 in the maturation of monocytes and macrophages in the gut mucosa. The authors use the well-described Cx3cr1-Cre-ERT-YFP mouse to specifically delete CD98 in this lineage. Collectively, the results show that loss of CD98 alters monocyte to macrophage differentiation and this results in diminished severity of chemically-induced colitis. These results will be of interest to the mucosal immunology field. However, there are major concerns that the authors need to address.

In Figure 1, the gate used to select CD64⁺ cells is very broad. The authors should show a staining control for the use of such a wide gating strategy.

We show now the respective staining control in supplementary fig. 1b.

Unbiased analysis using tSNE plots is a nice way to display these data. However, in order for the reader to properly interpret these, data showing the expression of each marker used to generate the tSNE analysis should be shown either in the main figure or as supplementary data. This is needed to understand which markers are generating the particular clusters shown in the BM and colon. The use of 'heatmap' tSNE are good for this purpose.

The surface markers that indicate the respective populations in the tSNE plots are indicated in the brackets of panel d and j of figure 1. We also generated a heatmap with 'Cytobank' to visualize what population expresses what marker. Moreover, t-SNE plots for each population is now presented in supplementary figure 1.

The overlay tSNE is difficult to visualise due to the rarity of e.g. P1. Also, given that so-called 'P5' cells dominate the macrophage compartment, why are grey cells not the dominant population on the tSNE plot? It would be more informative to have individual tSNE plots for each population. Having CD98 expression on the tSNE is rather meaningless given that most populations are uniformly positive.

Each population is now presented in individuals t-SNE plots in supplementary Fig. 1a (bone marrow cells) and Supplementary Fig. 1c (colonic lamina propria cells).

The identification of Kupffer cells (KC) is substandard. KC are the most abundant myeloid cell in the liver but in this manuscript, they form only a small population. Although KC are not the focus on the manuscript, conclusions are based on these data and it is not clear if these are true KC, especially because CD11c⁺ and CD11c⁻ subsets are identified which is not a conventional way to define these cells. In particular, the CD11c⁺ 'KC' do not fit the characterisation as KC given they lack high expression of F4/80 and low expression of CD11b. Another major concern is that the authors report an effect of tamoxifen

administration on CD98 expression amongst these populations in liver, which is unexpected given that bona fide KC (like Langerhans cells) do not express CX3CR1 in the adult (Yona et al. 2013, Immunity, Chorro et al. 2009, JEM). The authors should either repeat these experiments and use a refined isolation protocol or remove reference to KC and instead state that these data analyse liver myeloid cells, which will likely include hepatic macrophages. As the data stand, reference to KC is not justified.

We agree with the reviewer that our initial characterization of Kupffer cells (KC) was suboptimal. For the identification of KC, we have used the strategy proposed by David et al. ¹. This study describes two KC populations, the CD11c-negative F4/80⁺CD11b⁺Ly6c^{lo}MHC II⁺ cells and the F4/80⁺ CD11c⁺CD11b⁺Ly6c^{lo}, MHC II⁺ KCs. In agreement with the work of Yona et al. ², both CD11c⁻ and CD11c⁺ KCs have very low or lack CX3CR1 expression. We therefore re-analyzed our data and recognized that among the CD11c⁻ and CD11c⁺ cells are a substantial fraction of CX3CR1. In addition, we stained our cells for Zeb2, which is expressed by plasmacytoid dendritic cells (pDCs), conventional dendritic cells type 2 (cDC2) and macrophages including KCs ^{3,4}. Flow cytometry revealed that CX3CR1-negative CD11c⁻ and CD11c⁺ liver myeloid cells have high Zeb2 expression, whereas CX3CR1-positive CD11c⁻ and CD11c⁺ liver myeloid cells have lower Zeb2 expression. Thus, we consider the initial described CD11c⁻ and CD11c⁺ cells as liver myeloid cells and not as KC. We hence changed the text of the result section describing our findings of figure 2. Moreover, we revised the discussion to avoid conclusions that are not justified by our results.

The authors should comment on the fact that MHCII⁻ macrophages fail to delete CD98 during colitis. The authors also need to discuss what MHCII⁻ CD64⁺ cells are, how they relate to MHCII⁺ macrophages and the potential reasons why they might behave differently.

MHCII⁻ macrophages can mature into MHCII⁺ macrophages. Thus, it has been proposed that this cell population is the pre-stage of mature macrophages, and MHCII⁻ macrophages can be considered as “immature” macrophage ⁵. In this developmental stage, “immature” macrophages might express CX3CR1 in a lower extent which leads to an insufficient CD98hc deletion. Additionally, in figure 1g we show the morphology of the monocyte intermediates, and of the MHC II⁻ and MHC II⁺ macrophages.

In Figure 4 the order of the results does not follow the order of elements e.g. Fig. 4d is mentioned first. The data in Figure 4 needs to be presented in a more logical manner.

We have changed the order of results in figure 4, now figure 5.

The exact gating strategy for the sorting for scRNAseq should be shown. My concern about the broad CD64 gate is highlighted here, where many ‘contaminating’ clusters have been identified.

We added the gating strategy for cell sorting to Supplementary Fig. 6, panel a. Besides we have now performed complementary analyses to confirm the identity of each clusters and identify contaminant clusters (see below).

Confirmatory analysis must be done to show the transcriptional output matches the input (i.e. there is not selective sequencing of a particular subset or from a particular source). Expression of known markers e.g. CCR2, CD64, H2.Aa, CD74, Csf1r, CD68, Lys2 (Macrophages); Cd209a, Napsa (DC), S100A9 (neutrophils) etc would validate that all cells belong to the macrophage lineage. This would be best shown on tSNE plots for each marker. This is important given the authors identify 4 subsets that seem unrelated. The nature of these data allows identification of positive markers of these clusters (5, 7, 8 and 9). This data should be presented to provide justification for the exclusion of these subsets. scRNAseq provide unparalleled ability to identify novel subsets, therefore full justification for exclusion of subsets is essential.

We provide the projection of expression levels on the tSNE plots for all of the suggested markers. To further annotate our data set we used SingleR as suggested by reviewer # 2, and some enrichment test based on the overlap of the expressed genes as suggested by reviewer # 3, both using the immunological genome project (ImmGen) sorted bulk samples as reference⁶. After performing this analysis, it appears that cluster 5 was mainly composed of dendritic cells, cluster 7 of dendritic cells, monocytes and macrophages, cluster 8 of B cells and ILCs, and cluster 9 of fibroblasts.

The authors state in their introduction that Tim4 and CD4 expression can be used to identify long(er)-lived macrophages in the gut mucosa. To align these data with existing data in the field, the authors should show where CD4 and Tim4 align on their cluster analysis.

We searched in our data set for Cd4 and Timd4 expression. Cluster 3 and 6 (macrophages) and cluster 8 (B cells and ILCs) express CD4. Cluster 3 and 6 (macrophages) and cluster 5 (dendritic cells) express Timd4. This is now described in the result sections on page 12 (in the section: Single-cell RNA sequencing suggests a developmental trajectory of monocytes to macrophages in the colonic lamina propria).

As well as the merged tSNE in Fig. 4b, the same plots of just cells from tamoxifen treated versus corn oil treated mice need to be shown. This will allow the reader to assess if cells from each condition contribute to all of the clusters. A graph showing the contribution of WT vs CD98 KO cells to each cluster (e.g. SFig. 2f) should be shown in the main figure.

This was changed as suggested in Fig. 5b.

Can the authors confirm that all mice used for the scRNAseq analysis were of the same sex? This is important given that Xist is highlighted as a defining gene in Fig. 4a

Only females and littermates were used for the scRNAseq experiment. We state this in the method section as well as in the results sections, where the scRNAseq experiment is described. *Xist* (X-inactive specific transcript) is a RNA gene that plays an essential role in X-chromosome inactivation. However, in lymphocytes from females *Xist* can become activated

and can be detected in T cells, B cells, NK cells, dendritic cells and macrophages^{7, 8}. Thus, *Xist* is a hypervariable gene that can be expressed in immune cells of females and can therefore be detected in monocytes and macrophages of females.

The data in Supp. Figure 3c should not be shown as a line graph as these are not repeated measured from the same individuals. Also, given the variability in cell numbers from colonic digests, the data should always be compared with a control mouse colon digested on the same day. These data need to be added.

We agree with the reviewer that these are not repeated measures from the same individual mice, and therefore have changed the line graph to bar graphs. For every individual measure we have determined the cell numbers from colonic digests, which shows low variability between different colonic digests in our hands with no significant differences between different experiments. Since we have strict regulation of animal experimentation by the local authorities, that request the implementation of animal protection in sense of the three Rs whenever possible, we have decided not to use control animals for every individual data point in order to reduce the number of used animals. We hope that the reviewer is satisfied with our imperfect answer.

Ad Figure 7b

Ad Supplementary Figure 9c

FACS plots should accompany the data presented in Fig. 6B. The visual of disruption of the monocyte differentiation process will aid the reader.

FACS plots have been added to new Fig. 7d.

It is unclear what is being shown in Figure 6C. If it is a ratio (of KO/WT cells) of the proportion of each subset, then the exact details used to calculate this is need in the figure legend. Also, these data reveal that there is a 3-7 fold increase in the proportion of MHCII⁻ cells in the absence of CD98, which parallels the loss of MHCII⁺ macrophages. This could suggest that loss of CD98 changes the fate of monocytes to a MHCII⁻ macrophage rather than MHCII⁺ cell. Indeed, the absolute number of this subset is increased with tamoxifen treatment (i.e. CD98 deletion).

We have changed the figure legend of the former figure 6c (now figure 7d). Moreover, we agree with the reviewer that the fate of the monocytes could be changed that they develop to a MHCII⁻ macrophage rather to a MHCII⁺ macrophage. We have revised the text of the manuscript to appreciate the point raised by the reviewer

The data in Figure 7 would seem more appropriate in Figure 3.

We agree with the reviewer that the original figure 7 is more appropriate in figure 3. This figure appears now before the colitis experiments in mice.

That deletion of CD98 leads to attenuated colitis seems counterintuitive given that CCR2-dependent monocytes appear to drive chemical colitis and that these cells accumulate in CD98^{CX3CR1} mice. This needs to be discussed. Indeed, the discussion as a whole is too brief and fails to discuss much of the results.

We have revised and tried to improve the discussion, because we agree that the previous discussion was too brief. CCR2 facilitates the migration of monocytes in the inflamed lamina propria. Previous work has demonstrated that CCR2-deficient animals are somewhat protected from DSS colitis. For example, Andres et al. show that wt and CCR2^{-/-} mice have comparable clinical signs of colitis, but, however, showed reduced macroscopic and histological signs of inflammation compared to wt animals at day 7 of DSS colitis ⁹. Furthermore, Platt et al describe an inflammatory TLR2⁺ macrophages with low MHC II expression and high CCR2 expression ¹⁰. As consequence, in this study the lack of CCR2 in mice attenuated DSS colitis. Moreover, Blocking of CCR2 in animals led to reduced chronic colitis cancer associated with reduced numbers of infiltrating macrophages ¹¹. Furthermore, the depletion of monocytes with diphtheria toxin in CCR2 diphtheria toxin receptor transgenic (CCR2.DTR) protected from colitis ¹². Thus, this important point is now discussed on page 17.

The authors should state in each figure legend how many times a particular experiment has been performed to allow reproducibility to be assessed.

We provide now the information in the figure legends.

Reviewer #2 (Remarks to the Author):

This is a compelling report that addresses macrophage CD98 in intestine using Cx3cr1-CreER for inducible deletion of the heavy chain of CD98 in mice. This Cre proved to be usefully specific, because, as opposed to colonic lamina propria monocytes and macrophages, at least in the colon, non-myeloid lineages that express CD98 did not have decreased expression of CD98 with tamoxifen. Furthermore, the mice were protected from DSS-induced colitis with CD98 deletion in Cx3cr1-expressing cells. The expression of apoptosis pathway genes in KO cells found by single cell transcriptomics aligns well with the finding of fewer macrophages. Thus, the report advances knowledge of the role of CD98 in macrophage ontogeny in colitis, and the clinical data are of interest. Thus, the findings represent a noteworthy contribution. The major shortcoming is that the link between CD98 and protection from apoptosis is not made.

The authors mention data not shown that are quite intriguing: “Furthermore, feeding animals with 10% higher amino acid chow (5% more leucine and 5% more isoleucine) to induce an increased inflammation on a CD98 dependent manner, indicated a tendency to escalate DSS-induced inflammation (data not shown) compared to conventional chow. These results indicate that the loss of CD98 in CX3CR1⁺ macrophages attenuated DSS-induced colitis in mice.” It would be worthwhile to present these data and include them as a supplement, even if the significance is at a trend level. Were there more macrophages with extra amino acids?

We have added these data to the supplementary material (Supplementary Fig. 4) and describe these results on page 10. Significant differences in disease activity score, colon length and histological scores between groups of animals receiving regular chow and animals feed with a high amino acid diet was not observed with exception of disease activity scores in animals feed with high amino acid diet after conditional deletion of CD98hc. This could mean that the high amino acid diet may influence other cells then macrophages after conditional deletion of CD98hc or that macrophages somehow influence other cells. To further dissect this finding, we believe that this would extend the focus of our manuscript. In addition, we analyzed the numbers of CX3CR1⁺ macrophages in animals with colitis receiving a diet enriched with amino acids. We did not observe an increase of CX3CR1⁺ macrophages in colonic lamina propria of these animals compared to animals receiving regular chow. However, the conditional deletion of CD98hc resulted in reduced macrophage numbers confirming our findings obtained by flow cytometry.

In this connection, mTOR mediates inflammatory activation of macrophages (PMID:24280772, Byles et al. Nat Comm), in a manner that is mediated by transporter Slc7a5 (PMID 29422900, Yoon et al Frontiers in Immunology). In vitro or in vivo, could colonic macrophages with and without CD98 be treated with leucine with measurement of M1 markers and mTOR pathway activation?

We have determined the phosphorylation of the ribosomal protein S6 kinase beta-1 (p70S6K) and its target substrate S6 ribosomal protein (S6), which are both downstream of mTORC1. We did not observe difference in p70S6K and S6 phosphorylation after conditional deletion of CD98hc in mice with colitis (Supplementary Fig. 5a).

Moreover, we measured p70S6K and S6 phosphorylation in bone marrow derived macrophages (BMDM) stimulated with LPS + IFN- γ . The stimulation of LPS + IFN- γ induced p70S6K and S6 phosphorylation compared to non-stimulated BMDM and BMDM stimulated with IL-4 and IL-13. The supplementation of L-leucine to non-essential amino acid (NEAA) medium did not induce p70S6K and S6 phosphorylation. However, L-leucine supplementation to NEAA medium induced *Il-1 α* expression in BMDM. The stimulation of BMDM with LPS + IFN γ in EAA medium served as a positive control confirming that macrophage cultured under M1 conditions express *Tnf*, *iNos*, *Il6*, *Mcp1*, *Ilb*, *Kc* and *Il-1 α* (Supplementary Fig. 5b-d).

Furthermore, we cite now the important and interesting work of Byle et al. and the manuscript of Yoon et al. in the introduction.

In this manner, could a link between CD98, protection against apoptosis, and mTOR be made? In sum, some mechanistic insight into the anti-apoptotic pathways regulated by CD98 should be feasible within a short time frame and would a valuable addition to this story.

We faced the difficulty that tamoxifen-induced Cre-mediated recombination in *in vitro* cultures with BMDM or blocking the branched-chain amino acid transporter with D-phenylalanine resulted into the death of the cells (Supplementary Fig. 9). Thus, we were limited in further dissecting the molecular pathways how CD98hc prevents apoptosis. However, we measured p70S6K phosphorylation in macrophages *in vivo* after conditional deletion of CD98hc, where we did not observe a significant difference (Supplementary Fig. 9a).

The single cell data are intriguing but cellular annotation is marker-based. Existing free, publicly available annotation tools (for example, among several options: PMID: 30643263, Aran et al. Nat Imm) that use reference-based annotation for unbiased identification of cell types will confirm contaminating cell types and tighten the analysis of putative cell types of interest (monocytes and macrophages).

We used the tool described by Aran et al. to annotate the clusters of our data set by comparing them with publicly available reference data sets ⁶. We present these findings in the text and in supplementary Fig. 6c.

Reviewer #3 (Remarks to the Author):

The Branched-Chain Amino Acid Transporter CD98 Heavy Chain Facilitates the Development of Colonic Macrophages Associated with Decreased Apoptosis in Macrophage Progenitors

Philipp Wuggenig, Berna Kaya, Hassan Melhem, Korcan Ayata, Swiss IBD Cohort Investigators, Petr Hruz, Hideki Tsumura, Mориhiro Ito, Julien Roux, Jan Hendrik Niess

Summary

The study is aimed at investigating the importance of CD98 on monocyte differentiation into gut macrophages and implications on the pathogenesis of IBD. They show that mononuclear phagocytes and their progenitors express CD98hc in steady state as well as in inflamed conditions. They successfully established a mouse model, CD98hc^{ACX3CRI}, that silences the expression of CD98 specifically in cLP monocytes and macrophages upon tamoxifen injection and showed that loss of CD98 was associated with attenuated colitis in this model. The authors indicate that CD98 is required for the differentiation of Ly6C^{high} monocytes into gut macrophages.

Overall, the manuscript is original and relevant and addresses an important question of the immunological basis of IBD. The introduction however, should be reviewed to provide a clearer description of the model / rationale. The discussion section should also be reviewed and expanded to further discuss the findings presented and their significance; for instance, in the second paragraph of the discussion (lines 306-313) when the authors describe the impact of silencing CD98, their importance should be better contextualized. The discussion should also include a deeper discussion of the scRNAseq /developmental trajectory findings, including the flow cytometric assessments designed to verify the patterns identified in the scRNAseq analysis.

We have re-written and tried to improve the introduction and discussion of the manuscript. The scRNA-seq/developmental trajectory findings are now discussed on page 15.

Major Suggestions

1. The authors should further detail the “monocyte waterfall” in the introduction, including the importance of CX3CR1 as a marker. They should also refer to supplementary fig. 5 in the text and add the marker CCR2 in this schematic representation of the “monocyte waterfall” development (Supplementary Fig. 5).

We have revised and tried to improve the entire discussion to better introduce the development of colonic lamina propria macrophages through intermediates of the ‘monocyte waterfall’. In addition, we added the marker CCR2 to the scheme (now supplementary Fig. 11)

2. The selection of the CX3CR1CreER mice to generate the CD98hcKO mice should also be expanded in the introduction; also consider including a schematic figure to summarize the model as part of the supplementary material.

We have revised the entire introduction and discussion and state that colonic macrophages express CX3CR1. We also provide a scheme that explains the construction of CD98hc^{fllox/fllox} mice in supplementary figure 3 as described by¹³.

3. The authors mention in the introduction that long-lived lamina propria macrophages express Tim-4 and surface CD4, therefore they should include these markers in their analysis/discussion.

We describe this analysis now on page 12/13 of the result section,

4. Supplementary Fig. 2a: the frequency of CD98 was higher in B cells and neutrophils of CD98hc^{fllox/fllox} and CD98hc^{ΔCX3CR1} mice, respectively, than in their controls (corn oil). What are the authors thoughts on these observations?

We re-analyzed the data. The new figure shows a difference in DCs in KO tmx vs. corn oil and in NK cells wt tmx vs. corn oil. Explanation might be the effect of DSS-induced inflammation in combination of tamoxifen/corn oil treatment.

5. Supplementary Figure 2a: 10% of macrophages that remain after Tamoxifen treatment. Is this an indication that recombination was incomplete? Please clarify and include the gating strategy for each panel of this Figure (ie. Are the macrophages pre-gated on CX3CR1?).

We provide now this information in Supplementary Figure 3d. The macrophages are gated as CD64⁺ CX3CR1/YFP⁺. Approximately 90% silencing of CD98hc during tmx injection over five consecutive days on day 7 of the readout might be due to the high replenishment of colonic monocytes/macrophages by extravasated Ly6C^{high} monocytes.

6. The authors also state that feeding animals with 10% higher amino acid chow (5% more leucine and 5% more isoleucine) indicated a tendency to escalate DSS-induced inflammation (data not shown) compared to conventional chow. The results should be included as part of the supplementary material. What is the impact of this diet in the frequency/density of CX3CR1^{high} gut resident macrophages and the CXCR1^{int} mononuclear phagocytes?

We have now included this data into the supplementary material (supplementary fig. 4). High amino acid diet did not influence the frequency/density of CX3CR1 gut macrophages. However, in this experiment we could confirm that the conditional deletion of CD98hc results in reduced macrophage numbers in the colonic lamina propria.

7. Methods for liver perfusion: Please expand and add detail to the methods to include the dose of isoflurane, the rate of perfusion (ml/min), whether there was an in situ perfusion with pre-warmed media as the first step of the two step collagenase perfusion. Please detail the perfusion rate of step 2 (with the liver digest medium). Please fully describe the duration rate and temperature of the 5ml perfusion step.

We provide now this information in the methods.

8. Figure 4: To assess the quality of single-cell libraries, more sequencing details should be provided. Including the number of UMI detected per cell, the percentage of reads aligned to the transcriptomes, introns, ribosomes, and mitochondria, and the sequencing saturation. Please include 10x Genomics web summaries for each sample in the supplementary data. Please also include the ranked list showing the DE genes for each of the 9 clusters as supplementary data

More information is now included in the results, the 10x Genomics web summaries for each sample is now given in the supplementary material, and the list of DE genes per cluster is now in Supplementary Table 1.

9. Figure 4: For clarity, please show the sort strategy utilized prior to single cell RNA sequencing in the supplementary material. The authors state CCR2+/CD64+, does this mean either/or, or both.

The gating strategy prior scRNA-seq is now given in Supplementary Fig. 6a. The gate for sorting prior scRNA-seq included CCR2+CD64- / CCR2+CD64+ and CCR2-CD64+ cells.

10. Figure 4B: It is difficult to differentiate between control and Tamoxifen treated for the map of all 9 clusters. Please show a cluster map in which all control cells are one color and all Tamoxifen treated cells are a different color. It appears that there might be a higher frequency of cluster 2 cells in the KO vs the control mice. Please comment on any differences in the frequency of cells in each cluster for control vs KO.

We separated the respective tSNE plots showing the cluster map of control and cKO cells, and present this data in Fig. 5.

11. Figure 4B: Please annotate Clusters 5, 7, 8, and 9 did not seem to be logically related to other clusters using CIBERSORT or GSEA (Evaluation of methods to assign cell type labels to cell clusters from single-cell RNA-sequencing data JJ Diaz-Mejia et al., 1000Research, 2019).

We have tried to upload our data set on CIBERSORT but unfortunately failed to create a mouse-specific signature file (although no explicit error message was sent). We wrote to the authors but got no answer so far.

In parallel we tried two different approaches to perform this cell type annotation, SingleR (Sup. Fig 6c) and an enrichment test for the overlap of expressed genes with the reference ImmGen mouse microarray samples (Supplementary table 1). In both cases the results were very clear and allowed to identify the trajectory and contaminants clusters.

12. Figure 6A: Why do the author observe and increase in CD98+ MHC II^{neg} macrophages between days 14 and 28 of daily tamoxifen treatment?

We observed a reduction of MHC II⁺ and an increase of MHC II⁻ macrophages in the colonic lamina propria after conditional deletion of CD98hc. Possibly, the fate of monocytes could change from the development of MHC II⁺ to the development MHC II⁻ macrophages after loss of CD98hc. Thus, we have revised the text of the manuscript to more specifically state that the conditional depletion of CD98hc leads to a reduction of MHC II⁺ macrophages.

13. Figure 6C: It is unclear what is being shown in the figure itself or the legend. Please clarify and show the intra group vs inter group variation, with all data points and statistics (stated as a mean).

We revised the figure legend. We calculated the ratio of KO/WT total cells.

14. *Figure 7: Please also evaluate the CD98 expression in healthy individuals as a baseline comparator.*

We collected biopsies from healthy volunteers undergoing screening colonoscopy in our unit and determined CD98 expression. Furthermore, we stained for CD98hc in biopsies, and included these data into the former figure 7 (now figure 3).

Minor Issues

1. *Figure 7D is described in the figure legend but not displayed in Figure 7 as it has been moved to Supplementary Figure 5.*

This has been changed.

2. *Figure 1F: state the difference between P4 and P5 in the figure (MHCII^{neg} and MHCII^{pos});*

In addition to the flow cytometry data, we provide now scanning electron microscope images of the respective cell populations. The MHC II⁺ population has a more flattened surface structure compared to the MHC II⁻ macrophages. Possibly, the MHC II⁺ macrophages have cellular characteristics of more matured cells.

Further, Tamoutounour et al. describes the MHC II⁻ macrophages as ‘immature macrophages’ which subsequently mature into MHCII⁺ macrophages⁵.

3. *Line 229: refer to Supplementary Fig. 2d and e in the text;*

Appearance of figures in the text has changed.

4. *Line 273: refer to Fig. 6b in the text;*

In the revised version of the manuscript, we change the appearance of the figures.

5. *Line 44: followed “by”;*

We corrected this mistake. In general: new introduction

6. *Line 86: remove “in”;*

We apologize, corrected. In general: new introduction

7. *Line 346: Figure 7D is not shown and should be removed from the text (should it read Supplementary Fig. 5?).*

We changed the order of figures in the revised version of the manuscript.

8. *Figures should be formatted to reduce white space (particularly Figure 1).*

We tried to reduce white space within individual figures.

References

1. David, B.A. *et al.* Combination of Mass Cytometry and Imaging Analysis Reveals Origin, Location, and Functional Repopulation of Liver Myeloid Cells in Mice. *Gastroenterology* **151**, 1176-1191 (2016).
2. Yona, S. *et al.* Fate mapping reveals origins and dynamics of monocytes and tissue macrophages under homeostasis. *Immunity* **38**, 79-91 (2013).
3. Scott, C.L. *et al.* The Transcription Factor ZEB2 Is Required to Maintain the Tissue-Specific Identities of Macrophages. *Immunity* **49**, 312-325 e315 (2018).
4. Scott, C.L. & Omilusik, K.D. ZEBs: Novel Players in Immune Cell Development and Function. *Trends Immunol* **40**, 431-446 (2019).
5. Tamoutounour, S. *et al.* Origins and functional specialization of macrophages and of conventional and monocyte-derived dendritic cells in mouse skin. *Immunity* **39**, 925-938 (2013).
6. Aran, D. *et al.* Reference-based analysis of lung single-cell sequencing reveals a transitional profibrotic macrophage. *Nat Immunol* **20**, 163-172 (2019).
7. Wang, J. *et al.* Unusual maintenance of X chromosome inactivation predisposes female lymphocytes for increased expression from the inactive X. *Proc Natl Acad Sci U S A* **113**, E2029-2038 (2016).
8. Syrett, C.M. *et al.* Diversity of Epigenetic Features of the Inactive X-Chromosome in NK Cells, Dendritic Cells, and Macrophages. *Front Immunol* **9**, 3087 (2018).
9. Andres, P.G. *et al.* Mice with a selective deletion of the CC chemokine receptors 5 or 2 are protected from dextran sodium sulfate-mediated colitis: lack of CC chemokine receptor 5 expression results in a NK1.1+ lymphocyte-associated Th2-type immune response in the intestine. *J Immunol* **164**, 6303-6312 (2000).
10. Platt, A.M., Bain, C.C., Bordon, Y., Sester, D.P. & Mowat, A.M. An independent subset of TLR expressing CCR2-dependent macrophages promotes colonic inflammation. *J Immunol* **184**, 6843-6854 (2010).
11. Popivanova, B.K. *et al.* Blockade of a chemokine, CCL2, reduces chronic colitis-associated carcinogenesis in mice. *Cancer Res* **69**, 7884-7892 (2009).
12. Becker, F. *et al.* A Critical Role for Monocytes/Macrophages During Intestinal Inflammation-associated Lymphangiogenesis. *Inflamm Bowel Dis* **22**, 1326-1345 (2016).
13. Tsumura, H. *et al.* The role of CD98hc in mouse macrophage functions. *Cell Immunol* **276**, 128-134 (2012).

Reviewers' comments:

Reviewer #1 (Remarks to the Author):

The authors have addressed most of my initial comments, but I still feel that the manner in which they present their data is not very clear. Figures 1d,e,j and k do not provide any information that is not detailed in the preceding parts of the figure. Moreover, the heatmap analysis (e & k), is misleading as it suggests that Ly6Clow monocytes and P4 macrophages do not express CD98, which is evidently not the case. Showing the expression of markers used to define subsets is neither necessary nor informative.

Line 203-205 – This is not accurate. The authors have just shown that TAM leads to deletion in myeloid cells in the liver. Also, given the system relies on the Cx3cr1 promoter to drive Cre-ERT2, it would have been wise to examine another tissue that harbours Cx3CR1+ macrophages (such as the kidney, heart, brain) if the authors wanted to draw the conclusion that their system was specific to the gut.

Line 219 – it is difficult to draw the conclusion that the pattern of CD98 expression in human gut is equivalent to mouse cLP, as the murine data was at baseline and the human data is showing upregulation in IBD.

Line 234 – Typo – should read 'fed' rather than 'feed'

Line 236 – Typo – should read 'type' rather than 'typ'

Line 243 – Typo – should read 'fed' rather than 'feed'

Line 264 – Would suggest to change to 'we sorted all CD11b+ cells expressing either CCR2 or CD64'.

The data in Figure 5 have to be presented in a more logical format. It is rather confusing to jump directly to trajectory analysis without presenting the cluster analysis. This should be in the main figure as it is definitely not supplementary information, it is vital to interpret the figure. Clusters 1-9 are not defined anywhere. Clustering in Figure 5a does not match that in 5b, making it very difficult to know the identities of the clusters. From Figure 5b it looks like there are markedly fewer red events (can't determine the cluster number) but this is not commented on. While the data may support the conclusion stated on line 314, the current format of the paper make it impossible to tell.

The authors need to indicate at what point after TAM treatment were the CD98hcCX3CR1 mice used at for the scRNAseq. This is important because of the subsequent validation experiments showing major changes in the composition of the monocyte-macrophage compartment. Overall, the flow cytometric data do not support the conclusions made on the scRNAseq data that monocyte differentiation appears to be 'blocked'. Indeed, the data shown in Figure 7d support the idea that monocyte is altered, with some becoming MHCII- instead of normal MHCII+ macrophages. Of course, to show this definitively the authors would have to perform adoptive transfer of FACS-purified monocytes.

While this reviewer understands the constraints generated by 3Rs, the nature of tissue dissociation by enzymatic digestion means the cell yield can be variable. That controls are not included in each time point means firm conclusions on cell numbers must be made with caution. A line to this effect in the discussion should be added.

Line 1148 – Please clarify is whether 'trice' meant to read 'thrice' or 'twice'. Please check this for

accuracy – the figure legend states that there are 4 mice per group but only 3 data points are shown for the CD98hc-CX3CR1 mice. If one mouse was excluded, please state the rationale for this.

Reviewer #2 (Remarks to the Author):

The main conclusion of the paper remains sound, that CD98 is necessary for survival of colonic macrophages in the DSS colitis model, where they normally exacerbate disease. This finding is novel and should be of interest given the perspective it provides on macrophage ontogeny in colitis. The mechanism of enhanced apoptosis due to cKO, suggested by comparative transcriptomic analysis, is not clear; in this vein, the ancillary experiments done for revision, such as the in vivo amino acid supplementation or the in vitro assays with cytokine, are not conclusive and are somewhat disappointing. The authors should state this in the discussion. It is also not clear (and the authors should comment about this) why deletion of CD98 might actually increase disease score with high amino acid diet, and indeed why high amino acid diet as a lone variable has no effect on disease--in other words, stating that the data provide no evidence that the amino acid transport function of CD98 may have no relation to the observed effects would be valuable.

Reviewer #3 (Remarks to the Author):

This revision addressed many critical points. However, some of the additional information would be clearer if presented in a different manner.

I have the following specific comments that might help clarify the findings:

Major Comments:

1. In the rebuttal, the authors state "The surface markers that indicate the respective populations in the tSNE plots are indicated in the brackets of panel d and j of figure 1. We also generated a heatmap with 'Cytobank' to visualize what population expresses what marker. Moreover, t-SNE plots for each population is now presented in supplementary figure 1." Can the authors explain why the overall TSNE map for Figure 1d and supplementary figure 1a is different?
2. The new supplementary Figure 1 would be more informative for the readers if they also showed the expression of each marker that are used to delineate the populations (ie. In the case of supplementary Figure 1a (bone marrow cells), also show heatmaps displaying the levels of CD117, CD115, CD135, Ly6C, CD11b etc. In the case of supplementary Figure 1c (colonic lamina propria), also show levels of CD11b, CCR2 etc etc. An example of this type of representation of flow data is found in the paper by Niewold et al., Communications Biology volume 1, Article number: 227 (2018) (<https://www.nature.com/articles/s42003-018-0216-2/figures/1>).
3. Furthermore, new supplement 1b should be fully labeled: Please annotate the x and y axis of this plot and refer to the fact that it is a representative staining control for Figure 1f in the figure legend of the Supplementary Figure.
4. While the authors did mention that cd4 and timd4 were expressed in the dataset, they did not show that data, please include the heatmaps showing expression cd4 and timd4 in the Supplement.
5. Thank you for including the web summaries. Please provide information as to which of the 8 colons

and corresponding web summaries are for the 4 biological replicates of the corn oil or the 4 biological replicates from the tamoxifen treated animals. For the summary plots in 5b (control vs tamoxifen) how was batch correction carried out? While the data is appropriately normalized, the question of the impact of batch effect arises because the provided web summaries vary between 94 cells recovered per colon sampled (colon 3) to 696 cells recovered per colon sampled (colon 1), suggesting that there might be some expected variation in the colonic preps. Please show the representation of each of the 4 biological replicates that are making up the TSNE plots in the lower panels of 5b (what is the individual animal contribution to each cluster). The question is whether there is overlap in the samples, which would suggest that extensive batch correction might not be required. Please examine the animal/batch-specific effect in the data, regressing out technical factors including animal source, cells recovered, library size, and gene detection rate.

Minor typos:

- Please be consistent with italics for gene names
- Line 817: Do you mean 70 μm ?
- Line 1099: Trice
- Line 1148: Trice
- Line 1166: Exclusion

Point-to-point response to the reviewer's suggestions

(COMMSBIO-19-0371A)

Reviewer #1

The authors have addressed most of my initial comments, but I still feel that the manner in which they present their data is not very clear. Figures 1d,e,j and k do not provide any information that is not detailed in the preceding parts of the figure. Moreover, the heatmap analysis (e & k), is misleading as it suggests that Ly6Clow monocytes and P4 macrophages do not express CD98, which is evidently not the case. Showing the expression of markers used to define subsets is neither necessary nor informative.

We agree with the reviewer that the previous panels d,e,j and k do not give any preceding information of other parts of the figure. We have therefore removed the panels d,e,j and k from figure 1. As Reviewer # 3 has suggested to present tSNE plots to visualize the distribution of each marker used to identify the respective cell populations in a way for example published by Niewold et al., Communications Biology volume 1, Article number: 227 (2018) (<https://www.nature.com/articles/s42003-018-0216-2/figures/1>) we have therefore revised the supplementary figure 1. Supplementary figure 1 presents now tSNE plots showing the distribution of indicated markers used for the delineation of cell populations in the bone marrow and colonic lamina propria.

Line 203-205 – This is not accurate. The authors have just shown that TAM leads to deletion in myeloid cells in the liver. Also, given the system relies on the Cx3cr1 promoter to drive Cre-ERT2, it would have been wise to examine another tissue that harbours Cx3CR1+ macrophages (such as the kidney, heart, brain) if the authors wanted to draw the conclusion that their system was specific to the gut.

We agree with the reviewer that this statement is not accurate. We apologize that we have erroneously not changed the last sentence of this section in the previous revision cycle. We have now analyzed CD98hc expression by cardiac macrophages and microglia seven days after the first TAM injection. TAM treatment of CD98hc^{ΔCX3CR1} animals led to a reduction of CD98hc expression in cardiac macrophages but not in microglia. One possible explanation of our findings is, that cardiac macrophages are constantly replenished by extravasated blood monocytes in contrast to the microglia, which prenatally seeds the brain. Another explanation might be that the tamoxifen used in our study may not have sufficiently passed the brain-blood barrier. These data are presented in the new supplementary figure 4. Therefore, this statement was changed to “Overall, these results indicate that we have established a mouse model (CD98hc^{ΔCX3CR1}), in which tamoxifen injection successfully deletes the expression of CD98hc in a relatively selective manner in most CX3CR1/YFP macrophage populations”.

Line 219 – it is difficult to draw the conclusion that the pattern of CD98 expression in human gut is equivalent to mouse cLP, as the murine data was at baseline and the human data is showing upregulation in IBD.

We agree with the reviewer that this statement is inaccurate. Therefore, we changed the last sentence of this section to “Taken together, these data show that CD98hc and CD98lc are

expressed in the human cLP, and display higher expression in patients with quiescent and active IBD”.

Line 234 – Typo – should read ‘fed’ rather than ‘feed’

Line 236 – Typo – should read ‘type’ rather than ‘typ’

Line 243 – Typo – should read ‘fed’ rather than ‘feed’

Line 264 – Would suggest to change to ‘we sorted all CD11b+ cells expressing either CCR2 or CD64’.

The typos were corrected.

The data in Figure 5 have to be presented in a more logical format. It is rather confusing to jump directly to trajectory analysis without presenting the cluster analysis. This should be in the main figure as it is definitely not supplementary information, it is vital to interpret the figure. Clusters 1-9 are not defined anywhere. Clustering in Figure 5a does not match that in 5b, making it very difficult to know the identities of the clusters. From Figure 5b it looks like there are markedly fewer red events (can’t determine the cluster number) but this is not commented on. While the data may support the conclusion stated on line 314, the current format of the paper make it impossible to tell.

We have moved the cluster analysis from the supplement figures to the main figure 5, panel b and c. By comparing our data set with the data set provided by the ImmGen project we could annotate the cells depicted by our scRNAseq analysis. Cluster 1 are monocytes and macrophages, cluster 2 are monocytes, cluster 3 are macrophages, cluster 4 are monocytes, macrophages, endothelial cells, and epithelial cells, cluster 5 are DCs, cluster 6 are macrophages, cluster 7 are monocytes, macrophages, and DCs, cluster 8 are macrophages, innate lymphoid cells (ILCs), T cells, and B cells, and cluster 9 are fibroblasts and stromal cells. We also created tables summarizing the cluster analysis (supplementary tables 2 and 3)

In the previous revision cycle reviewer # 3 asked to present in addition to the tSNE plot showing WT and cKO cells in one plot, separate tSNE plots showing either WT or cKO cells. We apologize that we did not label the plots of the previous figure 5 sufficiently. We moved the tSNE plots showing either WT or cKO cells to supplementary figure 7, panel a. We also present tSNE plots, where we indicate the individual samples for each condition in supplementary figure 7, panel b. This is now described in the first paragraph of this result section.

We also revised the conclusion of this section to “Overall, these observations suggest that the excision of CD98hc in macrophages resulted in an altered ‘monocyte waterfall’-development to mature macrophages in the colonic lamina propria in tamoxifen-treated CD98hc^{ΔCX3CR1} mice”, ...

The authors need to indicate at what point after TAM treatment were the CD98hcCX3CR1 mice used at for the scRNAseq. This is important because of the subsequent validation experiments showing major changes in the composition of the monocyte-macrophage compartment. Overall, the flow cytometric data do not support the conclusions made on the scRNAseq data that monocyte differentiation appears to be ‘blocked’. Indeed, the data

shown in Figure 7d support the idea that monocyte is altered, with some becoming MHCII⁻ instead of normal MHCII⁺ macrophages. Of course, to show this definitively the authors would have to perform adoptive transfer of FACS-purified monocytes.

We carried out scRNAseq seven days after the first TAM injection (TAM treatment by daily injection for five consecutive days). This is now indicated on page 10 (Line 238 - 239).

Although we face constraints generated by 3Rs from the local authorities, we have repeated the experiment, in which colonic lamina propria (cLP) monocytes and macrophages at different time points after TAM injection were analyzed. In this new experiment we included for each time point littermates which were only treated with carrier (corn oil). We agree with the reviewer that the cell yield is variable between each isolation at different time points. We calculated the ratios between cell yields received from carrier or TAM-treated littermates for each cell population at the indicated cell points. This experiment shows that the MHCII⁻ macrophages are now also decreased but with less extent compared to the MHCII⁺ macrophages. Reasons for this decrease was an overall decrease in cell numbers but the relative proportion of MHCII⁻ macrophages increased. These new results are presented in the new figure 7.

Furthermore, we performed adoptive transfer experiments, in which CD45.1⁺ WT and CD45.2 YFP⁺ CD98hc^{ΔCX3CR1} bone marrow monocytes were transferred in CCR2^{-/-} recipients. We then calculated the ratios between recovered CD98hc^{ΔCX3CR1} (CD45.2⁺ YFP⁺) and CD45.1 WT normalized to input ratios in recipients receiving either TAM or corn oil (carrier). The CD45.2 YFP⁺ CD98hc^{ΔCX3CR1} / CD45.1 WT ratios was < 1 in TAM-treated animals with less extent for MHCII⁻ macrophages.

Moreover, we have revised our statement that “the monocyte differentiation is blocked” to “the monocyte differentiation is altered”.

While this reviewer understands the constraints generated by 3Rs, the nature of tissue dissociation by enzymatic digestion means the cell yield can be variable. That controls are not included in each time point means firm conclusions on cell numbers must be made with caution. A line to this effect in the discussion should be added.

As described above, we repeated the experiment where we included controls from carrier-treated CD98hc^{ΔCX3CR1} animals for each time point. This is presented in the new figure 7.

Line 1148 – Please clarify is whether ‘trice’ meant to read ‘thrice’ or ‘twice’. Please check this for accuracy – the figure legend states that there are 4 mice per group but only 3 data points are shown for the CD98hc-CX3CR1 mice. If one mouse was excluded, please state the rationale for this.

We checked and revised the figure legends.

Reviewer #2

The main conclusion of the paper remains sound, that CD98 is necessary for survival of colonic macrophages in the DSS colitis model, where they normally exacerbate disease. This finding is novel and should be of interest given the perspective it provides on macrophage ontogeny in colitis. The mechanism of enhanced apoptosis due to cKO, suggested by comparative transcriptomic analysis, is not clear; in this vein, the ancillary experiments done for revision, such as the in vivo amino acid supplementation or the in vitro assays with cytokine, are not conclusive and are somewhat disappointing. The authors should state this in the discussion. It is also not clear (and the authors should comment about this) why deletion of CD98 might actually increase disease score with high amino acid diet, and indeed why high amino acid diet as a lone variable has no effect on disease--in other words, stating that the data provide no evidence that the amino acid transport function of CD98 may have no relation to the observed effects would be valuable.

We agree with the reviewer that the experiments done for the revisions with amino acid supplementations are not conclusive and somewhat disappointing because these experiments did not give any further insights how on a molecular level the absence of CD98hc regulates apoptosis in macrophage progenitors. We also found increased disease scores with high amino acid diet in animals after deletion of CD98hc by macrophages. Potentially, other immune cells may compensate for the effects observed in cKO animals. To acknowledge this important point raised by the reviewer we have revised the discussion on page 18.

Reviewer #3

This revision addressed many critical points. However, some of the additional information would be clearer if presented in a different manner.

We thank the reviewer for the encouraging comment.

I have the following specific comments that might help clarify the findings:

Major Comments:

1. In the rebuttal, the authors state "The surface markers that indicate the respective populations in the tSNE plots are indicated in the brackets of panel d and j of figure 1. We also generated a heatmap with 'Cytobank' to visualize what population expresses what marker. Moreover, t-SNE plots for each population is now presented in supplementary figure 1." Can the authors explain why the overall TSNE map for Figure 1d and supplementary figure 1a is different?

tSNE is a machine-learning algorithm used stochastic neighbor embedding for visualization high dimensional data sets in two or three dimensions. It constructs a probability distribution, which explains that tSNE plots generated at different time points from the same data set look different. Because this reviewer has suggested to revise the previous supplemental figure 1,

we have generated new tSNE plots showing the distribution of the specific markers used for the identification of cell populations (see below). We have removed the tSNE and heatmaps from the main figure 1 for the sake of clarity that the presented tSNE do not look different generated at different times because of stochastic distribution of the data.

2. The new supplementary Figure 1 would be more informative for the readers if they also showed the expression of each marker that are used to delineate the populations (ie. In the case of supplementary Figure 1a (bone marrow cells), also show heatmaps displaying the levels of CD117, CD115, CD135, Ly6C, CD11b etc. In the case of supplementary Figure 1c (colonic lamina propria), also show levels of CD11b, CCR2 etc. etc. An example of this type of representation of flow data is found in the paper by Niewold et al., Communications Biology volume 1, Article number: 227 (2018) (<https://www.nature.com/articles/s42003-018-0216-2/figures/1>).

We have generated new tSNE plots for the new supplementary figure 1 showing the distribution of the individual markers used for the delineation of the respective cell populations in the bone marrow and colonic lamina propria.

3. Furthermore, new supplement 1b should be fully labeled: Please annotate the x and y axis of this plot and refer to the fact that it is a representative staining control for Figure 1f in the figure legend of the Supplementary Figure.

We apologize this mistake. The supplementary figure 1b is now fully labeled. The figure legend to supplementary figure 1b was changed to "representative staining control for Figure 1d", since the former panel f of figure 1 is now panel d.

4. While the authors did mention that cd4 and timd4 were expressed in the dataset, they did not show that data, please include the heatmaps showing expression cd4 and timd4 in the Supplement.

We presented Cd4 and Timd4 expression in the previous revision cycle in the tSNE plots of supplementary figure 7 and the heatmaps of panel d and panel e of the supplementary figure 6. In the revised version of the manuscript Cd4 and Timd4 expression is presented in figure 5 panel d, and tSNE plots of supplementary figure 8.

5. Thank you for including the web summaries. Please provide information as to which of the 8 colons and corresponding web summaries are for the 4 biological replicates of the corn oil or the 4 biological replicates from the tamoxifen treated animals. For the summary plots in 5b (control vs tamoxifen) how was batch correction carried out? While the data is appropriately normalized, the question of the impact of batch effect arises because the provided web summaries vary between 94 cells recovered per colon sampled (colon 3) to 696 cells recovered per colon sampled (colon 1), suggesting that there might be some expected variation in the colonic preps. Please show the representation of each of the 4 biological replicates that are making up the TSNE plots in the lower panels of 5b (what is the individual animal contribution to each cluster). The question is whether there is overlap in the samples, which would suggest that extensive batch correction might not be required. Please examine the animal/batch-specific effect in the data, regressing out technical factors including animal source, cells recovered, library size, and gene detection rate.

We have run all samples (4 controls, 4 cKO) for our scRNAseq experiment using littermate animals bred in the same facility in a carefully designed setup in parallel, excluding the possibility of the influence of different laboratory conditions, different reagents lots and different investigators on the sequence results. As suggested by the reviewer, we present tSNE plots, in which the individual animal source have been annotated for WT and cKO cells in supplementary figure 7, panel a. We found sufficient overlay between individual samples in cKO tSNE and control tSNE excluding batch effects in our in parallel performed experiment.

Minor typos:

-Please be consistent with italics for gene names

-Line 817: Do you mean 70 μ m?

-Line 1099: Trice

-Line 1148: Trice

-Line 1166: Exclusion

We have corrected these mistakes

REVIEWERS' COMMENTS:

Reviewer #1 (Remarks to the Author):

The authors have addressed all comments in their revised manuscript and their additional data makes the manuscript stronger.

Reviewer #3 (Remarks to the Author):

COMMSBIO-19-0371B-Z Wuggenig et al.,

The authors addressed all the outstanding concerns raised with additional experiments and figures.

Minor concerns:

New Supplementary Figure 1a shows TSNE graphs and states that the experiments were repeated 8 times. Were all 8 experiments included in the plots or is this representative? Please mention and expand the figures legend.

Point-to-point response to the reviewer's suggestions

(COMMSBIO-19-0371B-Z)

We thank the reviewers for the suggestions and insights to our manuscript.

Reviewer #1

The authors have addressed all comments in their revised manuscript and their additional data makes the manuscript stronger.

Reviewer #3

The authors addressed all the outstanding concerns raised with additional experiments and figures.

Minor remaining concerns:

New Supplementary Figure 1a shows TSNE graphs and states that the experiments were repeated 8 times. Were all 8 experiments included in the plots or is this representative? Please mention and expand the figures legend.

We choose one representative tSNE plot out of the eight performed experiments. We state this now in the legend to Supplementary Figure 1. Please note, that we have not marked the changes in the legends to supplementary figure in yellow since this is hopefully the last submission.